# Adenine base editing-mediated exon skipping restores dystrophin in humanized Duchenne mouse model

Jiajia Lin[1,6], Ming Jin[1,6], Dong Yang[2,6], Zhifang Li [3,6], Yu Zhang [2,6], Qingquan Xiao [2], Yin Wang[2], Yuyang Yu[2], Xiumei Zhang[2], Zhurui Shao[2], Linyu Shi [2], Shu Zhang[4], Wan-jin Chen[1], Ning Wang [1] ✉, Shiwen Wu[4] ✉, Hui Yang [2,5] ✉, Chunlong Xu [3,5] ✉ & Guoling Li [1,2] ✉

Duchenne muscular dystrophy (DMD) affecting 1 in 3500–5000 live male newborns is the frequently fatal genetic disease resulted from various mutations in *DMD* gene encoding dystrophin protein. About 70% of DMD-causing mutations are exon deletion leading to frameshift of open reading frame and dystrophin deficiency. To facilitate translating human DMD-targeting CRISPR therapeutics into patients, we herein establish a genetically humanized mouse model of DMD by replacing exon 50 and 51 of mouse *Dmd* gene with human exon 50 sequence. This humanized mouse model recapitulats patient's DMD phenotypes of dystrophin deficiency and muscle dysfunction. Furthermore, we target splicing sites in human exon 50 with adenine base editor to induce exon skipping and robustly restored dystrophin expression in heart, tibialis anterior and diaphragm muscles. Importantly, systemic delivery of base editor via adeno-associated virus in the humanized male mouse model improves the muscle function of DMD mice to the similar level of wildtype ones, indicating the therapeutic efficacy of base editing strategy in treating most of DMD types with exon deletion or point mutations via exon-skipping induction.

Duchenne muscular dystrophy is a fatal hereditary disease resulting from thousands of pathogenic mutations in human X chromosome-linked *DMD* gene[1]. It is estimated that 1 out of 3500–5000 live male newborns are affected by DMD, representing the most prevalent muscular disease. *DMD* gene encodes dystrophin protein involved in important cytoskeleton function and survival of body-wide muscle cells. Pathogenic *DMD* mutations were mostly distributed in some hotspot regions among 79 exons of *DMD* gene, leading to dystrophin loss and muscle atrophy[2]. Both point and deletion mutations could generate loss-of-function *DMD* alleles. About 70% DMD patients were affected by exon deletion mutations resulting in frameshift expression and loss-of-function of mutant dystrophin

protein[3]. However, there is still a lack of effective and curative therapy for DMD intervention.

Previous studies showed disruption of frameshift exon by Cas9 or other nuclease could restore functional dystrophin expression[4–7]. Nevertheless, nuclease-induced double-strand break (DSBs) not only generate a small fraction of in-frame and corrected *DMD* alleles, but also a large fraction of edited and uncorrected alleles carrying indel mutations without therapeutic effect[5–7]. In addition, large DNA rearrangements with detrimental effects might also be induced after DNA cleavage by Cas9[8–10]. Notably, most of the previous studies on gene editing treatment were not based on genetically humanized DMD mouse models[5–7,11–13], impeding their use of human-specific single

[1]Department of Neurology, First Affiliated Hospital, Fujian Medical University, Fuzhou, China. [2]HuidaGene Therapeutics Inc., Shanghai, China. [3]Lingang Laboratory, Shanghai, China. [4]Department of Neurology, First Medical Center of Chinese PLA General Hospital, Beijing, China. [5]Shanghai Center for Brain Science and Brain-Inspired Technology, Shanghai, China. [6]These authors contributed equally: Jiajia Lin, Ming Jin, Dong Yang, Zhifang Li, Yu Zhang. ✉e-mail: ningwang@fjmu.edu.cn; wu_shiwen@outlook.com; huiyang@huidagene.com; xucl@lglab.ac.cn; guolingli@huidagene.com

guide RNA (sgRNA) for testing therapeutic efficacy and off-target profile. Therefore, it would be necessary to directly edit and correct human *DMD* mutations with a highly accurate and safe gene editing tool.

Lately, adenine base editor (ABE) is becoming an attractive gene editing modality to correct genetic mutations by installing precise base conversion without double strand break[14-16]. To investigate the efficacy of ABE in treating the majority of DMD types caused by exon deletion mutations accounting for 70% of DMD patients, we herein generated a genetically humanized DMD mouse model by replacing mouse exon 50 and 51 with human *DMD* exon 50, modeling *DMD* deletion mutations and reliably recapitulating human DMD phenotypes. We then treated these humanized DMD models with ABEs to successfully restore body-wide dystrophin expression via inducing skipping of human exon 50. Importantly, DMD mice treated with systemic administration of ABE via adeno-associated virus (AAV) exhibited significantly improved muscle function, indicating the great potential of ABE in the therapeutic intervention of most common DMD types and other monogenic diseases.

## Results

### Establishment of the genetically humanized DMD mouse model carrying human exon 50 of *DMD* gene

To generate a genetically humanized DMD mouse model carrying human-specific exon deletion mutation, we knocked in the human exon 50 sequence to replace both exon 50 and 51 of mouse *Dmd* gene in a single step (Fig. 1a and Fig. S1a). The humanized male mouse model with the genotype of ΔE5051,KIhE50 was designated as DMD^ΔmE5051,KIhE50/Y afterwards. We used RT-PCR to examine the splicing product of *DMD* gene in wildtype (WT) and DMD^ΔmE5051,KIhE50/Y mice, showing smaller product size as predicted in DMD^ΔmE5051,KIhE50/Y than in wildtype mice (Fig. 1b). Sequencing results of RT-PCR product indicated proper splicing between mouse and human exons in DMD^ΔmE5051,KIhE50/Y mice (Fig. 1c).

To extensively characterize DMD^ΔmE5051,KIhE50/Y mice, tissues from 2-, 8- and 24-week-old mice were obtained and subjected to different analyses. We first performed immunostaining against dystrophin and spectrin proteins in heart, and diaphragm (DI) and tibialis anterior (TA). It found that DMD^ΔmE5051,KIhE50/Y mice exhibited complete loss of dystrophin expression in contrast to strong dystrophin expression in wildtype mice, but spectrin expression was unaffected in both wildtype and DMD^ΔmE5051,KIhE50/Y mice (Fig. 1d and Fig. S1b). Western blot results confirmed the dystrophin deficiency in DMD^ΔmE5051, KIhE50/Y mice (Fig. 1e). In addition, hematoxylin-eosin (H&E) and Sirius red histological staining revealed disorganized muscle fiber and severe fibrosis in TA and DI muscle but not heart tissue (Fig. 1f and Fig. S1c, d). We also measured creatine kinase (CK) activity in DMD^ΔmE5051,KIhE50/Y mice and found muscle damage-induced increase of serum CK activity after dystrophin loss (Fig. 1g and Fig. S1e). Consequently, DMD^ΔmE5051,KIhE50/Y mice showed significantly reduced levels of forelimb grip strength compared with wildtype mice (Fig. 1h and Fig. S1f).

These results demonstrated that the humanized DMD mouse model recapitulated patients' phenotypes, suggesting DMD^ΔmE5051,KIhE50/Y mice as a potential model to test human DMD-targeting CRISPR therapeutics.

### Local muscle administration of adenine base editor restores dystrophin expression

A recent study reported using Cas9 nuclease to disrupt exon 51 and restore dystrophin expression in another humanized DMD model[17]. Unlike Cas9 nuclease, base editor could install nucleobase conversion without induction of double-strand break in DNA to change RNA splicing outcomes. To investigate the potential of adenine base editing in treating DMD with our humanized mouse model, we designed 10 sgRNA to edit splicing regulation sequences around human exon 50 and tested their base-editing efficiency in HEK293T cells (Fig. 2a). It

showed detectable A-to-G editing activity for all 10 sgRNA but sgRNA6 with the highest efficiency of up to 50% (Fig. 2b). To investigate the efficacy of ABE for editing more *DMD* splicing acceptor (SA) and donor (SD) sites (Fig. S2a), we selected other frequently mutated exons than exon 50 in human *DMD* gene for evaluation. A total of 71 sgRNA was designed against SA and SD sites of hotspot exon 2, 43, 44, 45, 46, 51, 52, 53 and 55. We found detectable A-to-G editing for all tested sgRNA (Fig. S2b–j), suggesting the general applicability of editing splicing acceptor (SA) and donor (SD) sites via ABE for potential DMD intervention.

Since the adenine base editor has vector size beyond the genome packaging capacity of AAV, we have to split the adenine base editor into two fragments expressed by two separate AAV (Fig. 2d). Two base editor fragments were seamlessly spliced by intein enzyme to form the full-length base editor (Fig. S3a). Different intein enzymes were tested to achieve maximal base editing activity. *Rhodothermus marinus* (Rma) intein outperformed four other enzymes in generating full-length base editor to induce 50% of A-to-G editing rate achieved by the unsplit base editor (Fig. 2c). In addition, the optimal split position in the base editor was between 573 and 574 amino acid (a.a.) residue (Fig. 2c and Fig. S3b–d).

To test in vivo activity of two split base editors, ABE1 (with two sgRNA6) and ABE2 (with three sgRNA6) (Fig. 2d), we performed local injection of AAV-ABE in TA muscle of 3-week-old DMD^ΔmE5051,KIhE50/Y mice (Fig. 3a). At six-week after injection, TA muscle was collected for subsequent analysis (Fig. 3a). We found AAV-ABE2 exhibited over 15% of A-to-G editing rate greater than that of AAV-ABE1 (Fig. 3b). RT-PCR results indicated successful splicing change to skip human exon 50 after ABE-induced base conversion as confirmed by sequencing results (Fig. 3c, d). The exon skipping efficiency reached over 70% and 90% for AAV-ABE1 and -ABE2, respectively (Fig. 3e). To evaluate the off-target effect of ABE guided by sgRNA6, we predicted and analyzed potential off-target loci via sequence similarity analysis with Cas-OFFinder[18] and deep sequencing, showing high on-target editing with sgRNA6 but undetectable off-target editing for all 14 potential off-target loci with no more than three mis-match nucleotides with the target sequence (Fig. S4). For target site analysis, we found bystander editing near the on-target adenine (A7) for both ABE1 and ABE2 after in vivo intramuscular delivery via AAV (Fig. S5). However, bystander edits were predicted to not generate alternative splicing sites and cause no amino acid change in the exon50-skipping transcripts.

Furthermore, dystrophin expression shown by immunostaining results was remarkably rescued by local injection of AAV-ABE (Fig. 3f; Fig. S6). The percentage of dystrophin-positive fibers after ABE2 treatment reached the wild-type level. (Fig. 3g). To quantify dystrophin restoration level, we performed western blot analysis showing that 48.44 ± 18.05% and 96.32 ± 3.93% dystrophin was restored by ABE1 and ABE2 respectively (Fig. 3h, i).

### Intraperitoneal injection of adenine base editor in neonatal DMD mice rescues dystrophin deficiency and muscle function

Since dystrophin deficiency affected body-wide muscle function, we next performed intraperitoneal (IP) injection of AAV-ABE2 in neonatal DMD^ΔmE5051,KIhE50/Y mice to achieve systemic delivery of ABE vectors (Fig. 4a). Therapeutic efficacy was analyzed six weeks post injection for heart, DI and TA muscles (Fig. 4a). We found that AAV-ABE2 showed the highest A-to-G editing rate of 16.41 ± 1.84% in heart, surpassing the efficiencies observed in both the DI and TA muscles. (Fig. 4b). In addition, exon skipping efficiency was close to 100% in heart and about 50% in DI and TA shown by the RT-PCR and sequencing results (Fig. 4c, d).

We further performed quantitative western blot assay to show 78.38 ± 12.79% of dystrophin expression restored in heart and 54.74 ± 10.51%, 58.21 ± 9.79% in DI and TA, respectively, after AAV-ABE2 treatment (Fig. 4e, Fig. S7a). Immunostaining results also detected

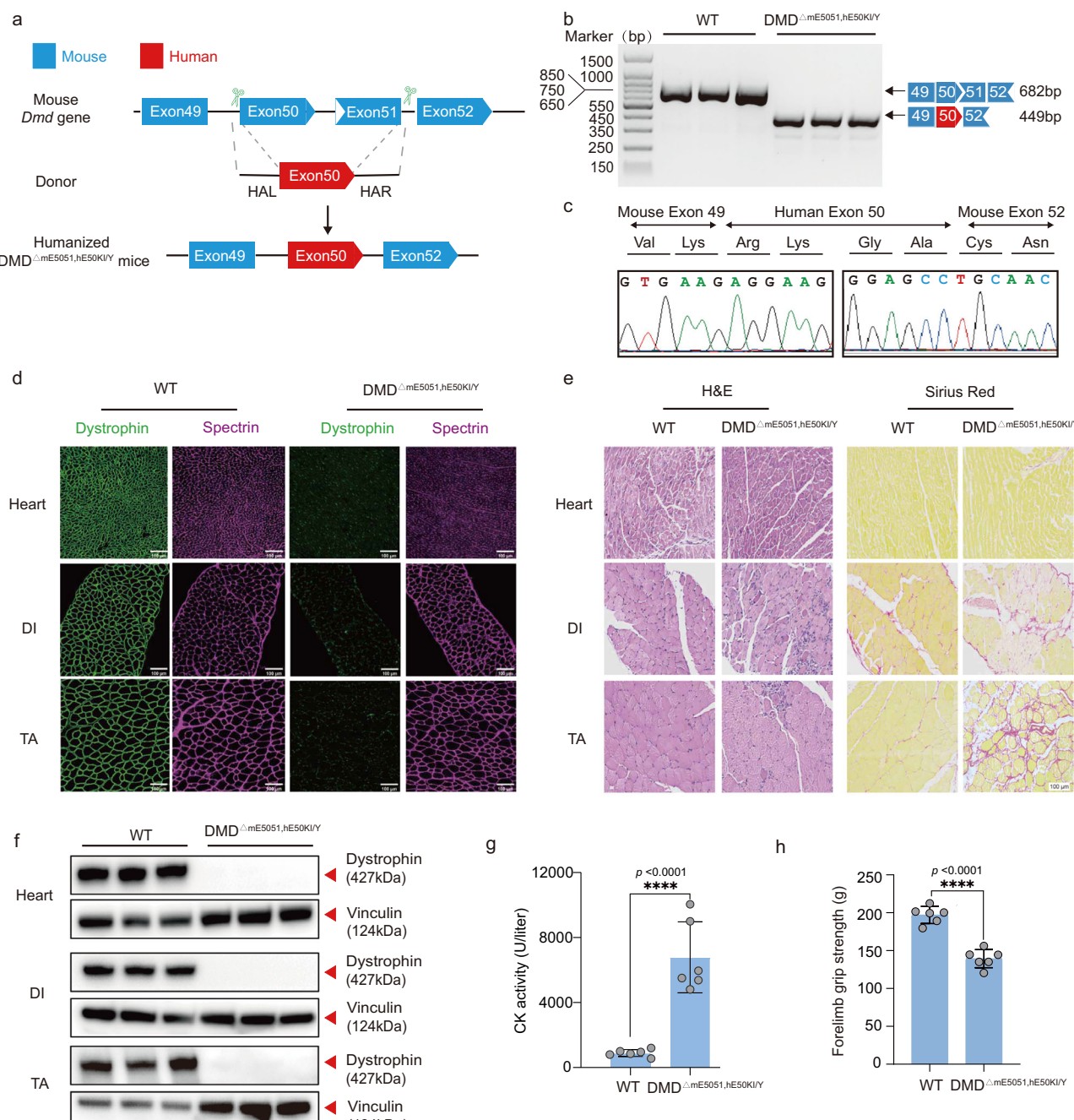

**Fig. 1 | Establishment and characterization of a humanized DMD mouse model.**
**a** Strategy for generating humanized DMD mouse model. Shape and color of boxes of exons indicate reading frame. CRISPR-Cas9 editing using two sgRNAs flanking exon50 and exon51 was used to delete mouse *Dmd* exon 50 and exon51 (blue box), and replaced with human *DMD* exon 50 (red box) flanking 200 bp intron sequence. **b** RT-PCR products from muscle of DMD$^{\Delta mE5051,KlhE50/Y}$ mice were gel electrophoresis and sequenced (**c**) to validate exon knock in and deletion. **d** Dystrophin immuno-histochemistry from indicated muscles of WT and DMD$^{\Delta mE5051,KlhE50/Y}$ mice. Dystro-phin and spectrin are shown in green and mangenta, respectively. **e** Western blot confirming the absence of dystrophin in indicated muscle tissues. **f** Hematoxylin-eosin (H&E) and Sirius red staining of tibialis anterior (TA), diaphragm (DI), and heart muscle of wild-type (WT) and DMD$^{\Delta mE5051,KlhE50/Y}$ mice. **g** Serum creatine kinase (CK), a marker of muscle damage and membrane leakage, was measured in WT and DMD$^{\Delta mE5051,KlhE50/Y}$ mice. **h** WT and DMD$^{\Delta mE5051,KlhE50/Y}$ mice were subjected to forelimb grip strength testing to measure muscle performance. All mice were 8 weeks old at the time of the experiment. Data are presented as mean ± s.d ($n = 6$ independent biological replicates). Each dot represents an individual mouse. Significance is indicated by asterisk and determined using unpaired two-tailed Student's *t* test. ****$P < 0.0001$. Scale bar, 100 μm. Source data are provided as a Source Data file.

strong expression of dystrophin in treated DMD$^{\Delta mE5051,KlhE50/Y}$ mice in contrast to undetectable signal in untreated DMD$^{\Delta mE5051,KlhE50/Y}$ mice (Fig. 4f). We found more than 85% of muscle fibers to be dystrophin positive for heart, TA and DI tissues after IP injection of AAV-ABE1 and ABE2 (Figs. S7b, S8). Furthermore, creatine kinase activity in the serum of treated mice was significantly reduced in contrast to untreated animals, suggesting decreased muscle damage after

ABE treatment (Fig. S9a). In addition, we also performed quantitative assay for two serum biomarkers, alanine aminotransferase (ALT) and blood urea nitrogen (BUN). ALT level was significantly lower in ABE-treated DMD$^{\Delta mE5051,KlhE50/Y}$ mice than non-treated ones (Fig. S9b), but BUN level remained unchanged before and after treatment (Fig. S9c), indicating the relative safety profile of AAV-ABE administration.

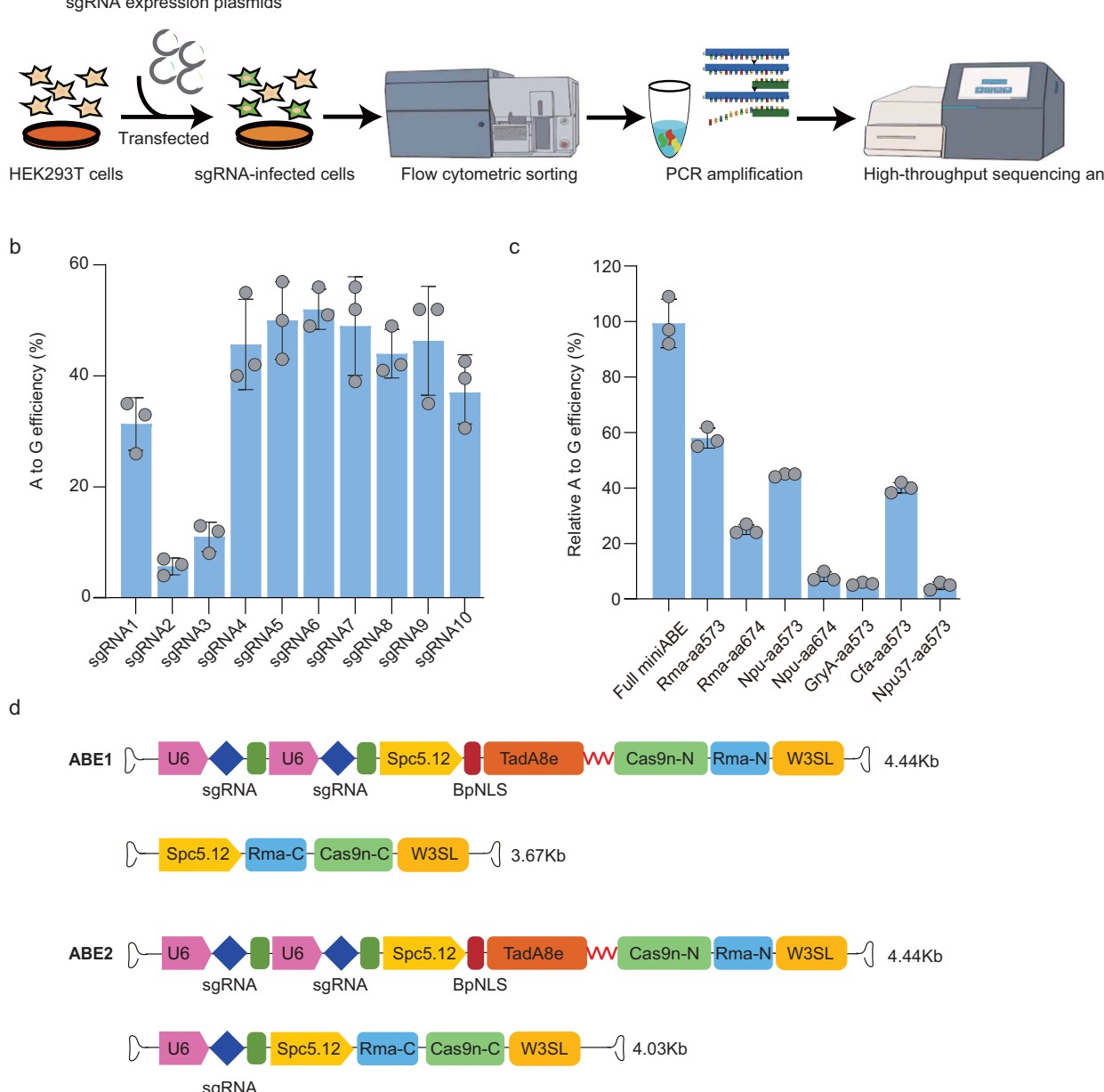

**Fig. 2 | ABE system efficiency mediated splice site mutation in vitro. a** The sgRNA screening scheme of adenine base editor (ABE) mediated editing of splicing site in HEK293T cells. **b** A-to-G conversion rate for 10 sgRNAs targeting exon 50 of *DMD* gene. **c** Comparison of the editing efficiencies for ABE systems split with various intein sequences. **d** Schematic of the AAV vectors for ABE1 and ABE2 variants. TadA8e, Cas9n split by *Rhodothermus marinus* (Rma) intein and the sgRNA6 were packaged into 2 separated AAV particles. A muscle-specific promoter Spc5-12 was used to drive TadA8e-Cas9n-N or Cas9n-C. ABE1 only contained two copies of sgRNA in Cas9n-C, but ABE2 contained two and one copies in Cas9n-C and TadA8e-Cas9n-N, respectively. Each dot represents individual biological replicates. Data are presented as mean ± s.d (*n* = 3 independent biological replicates). Significance is indicated by asterisk and determined using unpaired two-tailed Student's *t* test. Source data are provided as a Source Data file.

To examine the muscle function post treatment, we performed rotarod running test and measurement of forelimb grip strength. Consistent with dystrophin restoration, rotarod running time was significantly improved after AAV-ABE2 treatment compared with non-treated DMD$^{\Delta mE50I,KIhE50/Y}$ mice (Fig. 4g). Besides, forelimb grip strength was rescued to wildtype level in ABE-treated DMD$^{\Delta mE50I,KIhE50/Y}$ mice (Fig. 4h). Notably, grip strength durability was enhanced in ABE-treated DMD$^{\Delta mE50I,KIhE50/Y}$ mice as indicated by repetitive grip strength assay (Fig. 4i).

## Intravenous infusion of adenine base editor in humanized DMD mice induces restoration of dystrophin expression and muscle function

Clinically, intravenous infusion is commonly used to deliver medications and fluids to DMD patients. To closely reflect this clinical phenomenon, we decided to systemically administrated AAV-ABE in 2-week-old DMD mice for therapeutic efficacy analysis (Fig. 5a). We then kept injected DMD$^{\Delta mE50I,KIhE50/Y}$ mice via intravenous (IV) injection of AAV-ABE2 for treatment outcomes characterization at 6-week and 10-

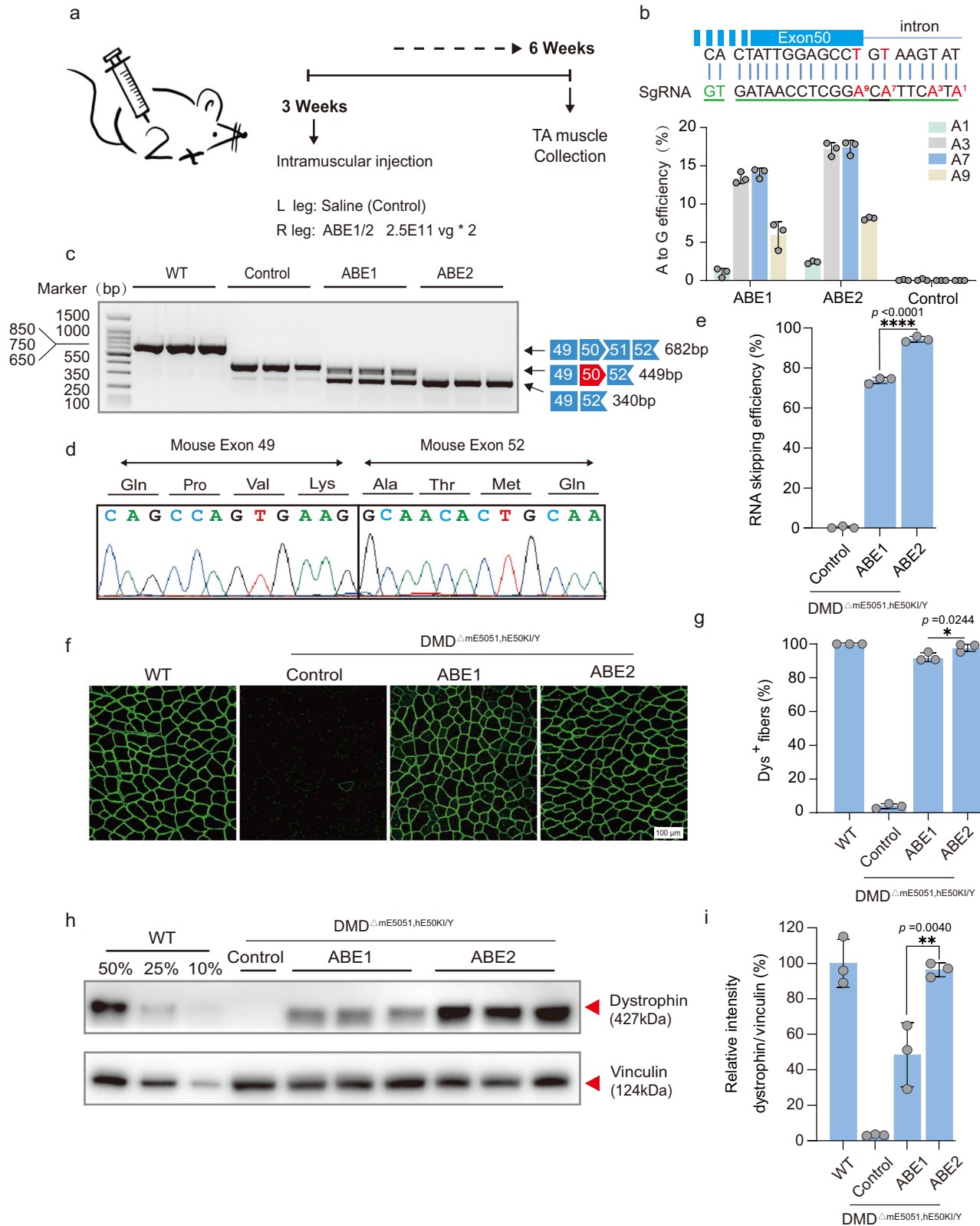

month post AAV-ABE2 injection (Fig. 5a). It showed 20.84 ± 4.8%, 4.34 ± 1.47% and 6.59 ± 2.17% A-to-G editing rate for heart, DI and TA muscle, respectively (Fig. 5b), which is similar to what found in neonatal treated mice. Furthermore, we found similar base editing outcomes in muscle stem cells by isolating Pax3 and Pax7 positive satellite cells for analysis at 6-week after tail-vein injection of AAV-ABE2 (Fig. S10a–c). RT-PCR analysis revealed the highest exon-skipping

efficiency in heart, two times higher than that in DI and TA (Fig. 5c and Fig. S11). Quantitative western blot assay showed more than 75% of restored dystrophin protein in heart after systemic AAV-ABE2 treatment while 50% of dystrophin restored in DI and TA (Fig. 5d and Fig. S12a). Moreover, dystrophin was readily detected by immunostaining in three muscle tissues after 6-week and 10-month treatment (Fig. 5e). There are more than 95% dystrophin positive muscle fibers

**Fig. 3 | ABE systems robustly rescued dystrophin expression in TA 6 weeks after local AAV injection. a** Overview for the *in vivo* intramuscular injection of the adeno-associated virus (AAV) -ABE construct into the tibialis anterior muscle of the right leg of 3-week-old DMD$^{\Delta mE5051,KlhE50/Y}$ mice. Left leg was injected with saline as a control. Black arrows indicate time points for tissue collection after injection. **b** The base editing efficiency of ABE1 and ABE2 were analyzed by deep sequencing. Sequence shows sgRNA (green line) and PAM (green letters). **c** RT-PCR products from muscle of DMD$^{\Delta mE5051,KlhE50/Y}$ mice were analyzed by gel electrophoresis. **d** RNA exon skipping was validated by Sanger sequencing. **e** Exon skipping efficiency by analyzing of total mRNA extracted from muscle tissue with deep sequencing. **f** Comparison of dystrophin (Dys+) expression restored by ABE1 and ABE2 systems

with immunofluorescence staining. Dystrophin is shown in green. Scale bar, 100 μm. **g** Quantification of Dys+ fibers in cross sections of TA muscles. **h** Western blot analysis of dystrophin and vinculin expression in TA muscles 6 weeks after injection with ABEs or saline. **i** Quantification of dystrophin expression from Western blots after normalization to vinculin expression. Age-matched wild-type (WT) and saline-treated DMD$^{\Delta mE5051,KlhE50/Y}$ mice were included as positive and negative control Data are presented as mean ± s.d ($n = 3$ independent biological replicates). Each dot represents an individual mouse. Significance is indicated by asterisk and determined using unpaired two-tailed Student's $t$ test, $*P < 0.05$. $**P < 0.01$. $****P < 0.0001$, NS represents not statistically significant. Source data are provided as a Source Data file.

rescued by AAV-ABE2 treatment in heart and TA, while more than 75% in DI muscles (Figs. S12b and S13). Furthermore, we found the muscle fibrosis and blood creatine kinase activity were dramatically alleviated by ABE treatment as indicated by H&E histology staining and CK analysis results (Figs. S14, S15a). In addition, serum ALT was also normalized to the wildtype level after ABE treatment while BUN remained the same before or after ABE administration (Fig. S15b, c), suggesting the safe and therapeutic outcomes after systemic delivery of AAV-ABE in 2-week-old DMD mice.

To interrogate functional recovery of ABE-treated muscle, we first measured rotarod running time and forelimb grip strength in ABE-treated versus untreated DMD$^{\Delta mE5051,KlhE50/Y}$ mice. It found significant improvement in terms of rotarod running time and forelimb grip strength after 6-week and 10-month ABE administration (Fig. 5f, g). Next, cardiac function was evaluated by echocardiography for untreated and treated DMD mice at 6 weeks and 10 months. Consistent with the previous study that *Dmd* mutant mice carried no cardiac defects due to the compensation of utrophin expression for dystrophin loss in the heart[19], we detected no difference in several electrocardiogram indexes among wildtype, treated and untreated DMD mice (Fig. S16). Notably, we performed repetitive grip test to reveal that ABE-treated DMD$^{\Delta mE5051,KlhE50/Y}$ mice outperformed non-treated ones in this assay (Fig. 5h), suggesting muscle function rescue with systemic AAV-ABE delivery in both neonatal and 2-week-old DMD mice.

## Discussion

To facilitate translational application of CRISPR therapeutics in clinics, it would be highly necessary to generate genetically humanized models carrying the same mutant sequences as in DMD patients. Previously, most of study on developing CRISPR-based DMD treatment performed gene editing of mouse DMD sequences[20], which shows variations between species and impedes the evaluation of human-specific sgRNA for translational study. Olson et al. and other groups recently created several humanized DMD mouse models carrying human exon 51, 45 deletion or other mutations, and showed restoring dystrophin expression via Cas9 cleavage or RNA editing of mutant human exons in these DMD models[17,21,22]. However, human *DMD* mutations occurring in several hotspot exons included but not limited to exon 51 or 45. Therefore, it warrants the establishment of other humanized DMD models to cover frequently mutated human exons. In previous studies, we created two nonsense mutation DMD mouse models incorporating exons 23 or 30 of the human dystrophin gene. Through a single systemic administration of ABE with human sgRNA, we were able to initiate widespread dystrophin expression throughout the body and enhance muscle function[23]. However, it is worth noting that specific exon nonsense mutations are rare among DMD patients[3]. In the regard, our present study demonstrated the successful one-step replacement of mouse exon 50 and 51 with human exon 50 to generate a humanized DMD mouse model for testing base editing treatment. This advancement holds promise for the majority of DMD patients, offering significant clinical benefits. Moreover, we also evaluated the efficacy of adenine base editor in editing of splicing donor (SD) and

acceptor (SA) sequences for other common exons affected in DMD patients, showing the general applicability of ABE-mediated SD and SA editing in DMD treatment. Recently, Chemello et al.[12] and other studies[24–26] reported using base and prime editors for splicing modulation and dystrophin restoration in patient-derived cardiomyocytes or in tibialis anterior muscle of a common *mdx* mouse model without genetic humanization via local AAV injection. However, our finding extensively complemented Chemello et al. study by demonstrating the applicability of base editing strategy for robust dystrophin rescue in genetically humanized *mdx* models over a long-term (10-month) observation period with the systemic AAV-ABE delivery method that is also similarly required for disease intervention in DMD patients.

Unlike the previous Cas9 cleavage strategy for *DMD* gene editing[17,21,27–30], we took adenine base editor to modify the splicing modulation sequence for human exon 50 skipping and restored the remarkable level of dystrophin expression in body-wide muscle tissues. For mutation disruption strategy, Cas9 nuclease induces double-strand break to not only generate therapeutic *DMD* alleles via mutant exon skipping or reframing, but also a large fraction of random indel alleles without therapeutic effect. In contrast, adenine base editor could efficiently restore dystrophin expression with no DSB induction. In addition, Cas9-treated gene may also introduce random repaired sequences around the cleavage site, which potentially produce proteins carrying neoantigen with immunogenicity risk. Thus, the ABE strategy shown in our present study and other studies[12,24] could avoid the above shortcomings and might represent a better alternative than cleavage-inducing strategy for DMD treatment.

Our study demonstrated that systemic administration of ABE in 2-week-old mice could also significantly restore dystrophin expression and alleviate muscle atrophy, which potentially would be helpful for the majority of patients in clinics already with DMD manifestation. Previous studies have shown that increasing the copy number of sgRNA can enhance gene editing efficiency[31,32]. To increase the in vivo editing efficiency of ABE, we designed three copies of sgRNA expression units in our AAV vector, exhibiting dramatic improvement of editing rate and restored dystrophin level for this optimized system compared to that of two gRNA expression unit. It is important to note the limitations associated with our sample size and the preliminary nature of our results. Subsequent studies with increased sample sizes are necessary to validate these findings and ensure the accuracy of our conclusions.

Modest DNA editing can achieve significant RNA restoration efficiency, as supported by similar findings in other studies[7,33–36]. The discrepancy is likely due to the presence of multiple cell types within the muscle tissue, including endothelial cells, pericytes, macrophages, fibro-adipogenic progenitors, and potentially other cell types that are not yet well understood at this time[37]. Additionally, nonsense-mediated decay (NMD) may influence the abundance of non-edited cDNA products, resulting in an apparent higher proportion of edited cDNA[38]. It is worth noting that, given the multinucleated nature of many murine muscles, even editing a single nucleus in a cardiomyocyte or skeletal muscles would result in the entire multinucleated cell

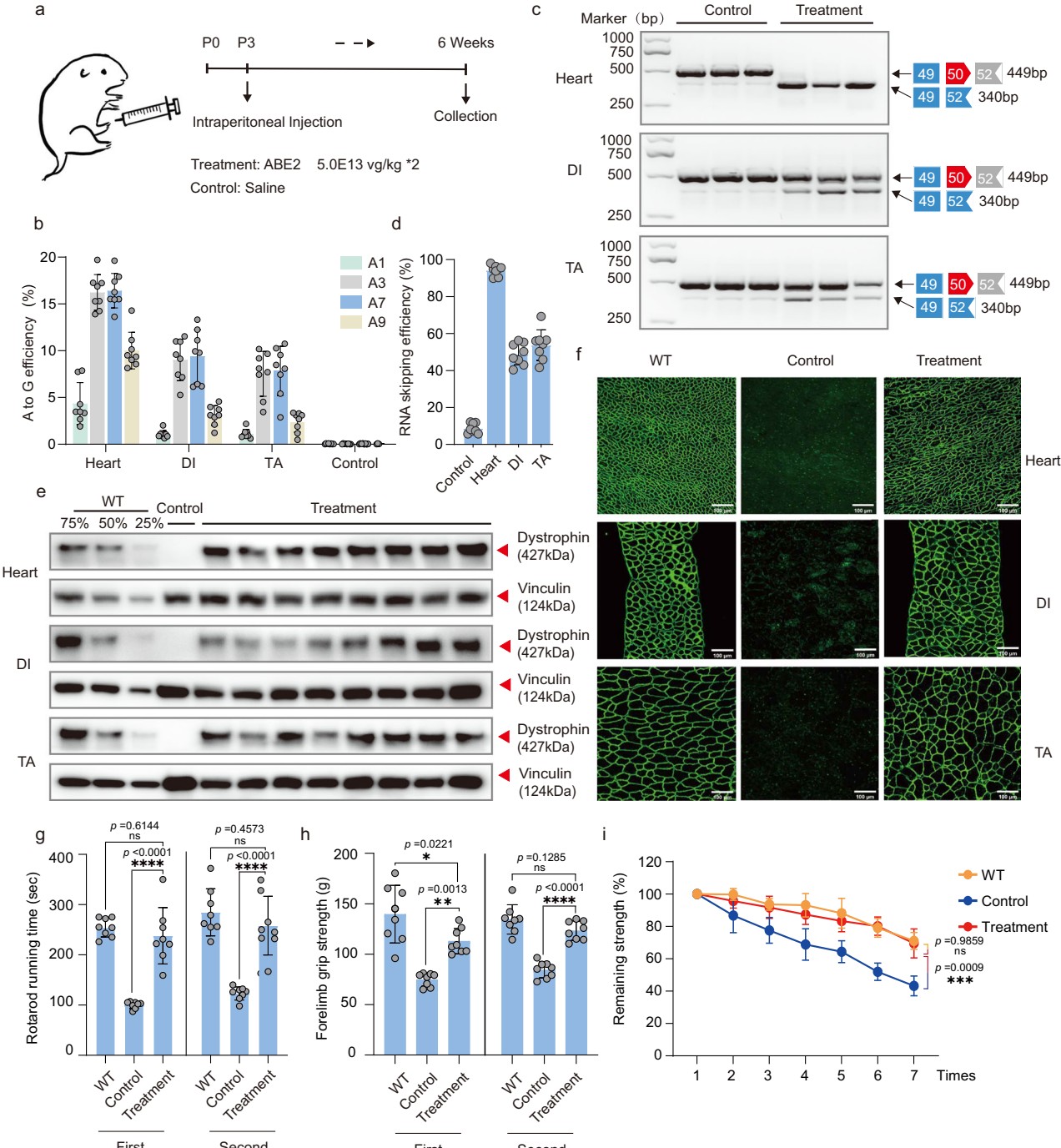

**Fig. 4 | Intraperitoneal systemic delivery of ABE system rescues dystrophin expression and muscle function in neonatal mice. a** Schematic of intraperitoneal injection of AAV particles. AAV-ABE2 was injected intraperitoneally (IP) into postnatal-day-3 (P3) DMD$^{\Delta mE50S1,KlhE50/Y}$ mice. Some DMD$^{\Delta mE50S1,KlhE50/Y}$ mice were injected with saline as mock-treated controls. Black arrows indicate time points for tissue collection after IP injection. **b** Measurement by deep sequencing of splicing site editing efficiency in tibialis anterior (TA), diaphragm (DI), and heart after systemic delivery. RT-PCR products from muscle of DMD$^{\Delta mE50S1,KlhE50/Y}$ mice were analyzed by gel electrophoresis (**c**) and deep sequencing (**d**) to validate exon skipping efficiency. **e** Western blot analysis shows restoration of dystrophin expression in the TA, DI, and heart of DMD$^{\Delta mE50S1,KlhE50/Y}$ mice 6 weeks after injection. Dilutions of protein extract from WT mice were used to standardize dystrophin expression

(25%, 50% and 75%). Vinculin was used as the loading control. **f** Immunohistochemistry for dystrophin in TA, DI, and heart of DMD$^{\Delta mE50S1,KlhE50/Y}$ mice was performed 6 weeks after IP injection. Dystrophin is shown in green. Scale bar, 100 µm. Rotarod rod running time (**g**) and forelimb grip strength (**h**) was measured two days in WT, DMD$^{\Delta mE50S1,KlhE50/Y}$ mice, and DMD$^{\Delta mE50S1,KlhE50/Y}$ mice treated with ABE2 particles. **i** The remaining strength was also measured during 7 repetitions test at 10-s intervals. Each dot represents an individual mouse. Data are presented as mean ± s.d ($n = 8$ independent biological replicates). Significance is indicated by an asterisk and determined using the one-way ANOVA multiple comparison test. *$P < 0.05$. **$P < 0.01$. ***$P < 0.001$. ****$P < 0.0001$, Ns represents not statistically significant. Source data are provided as a Source Data file.

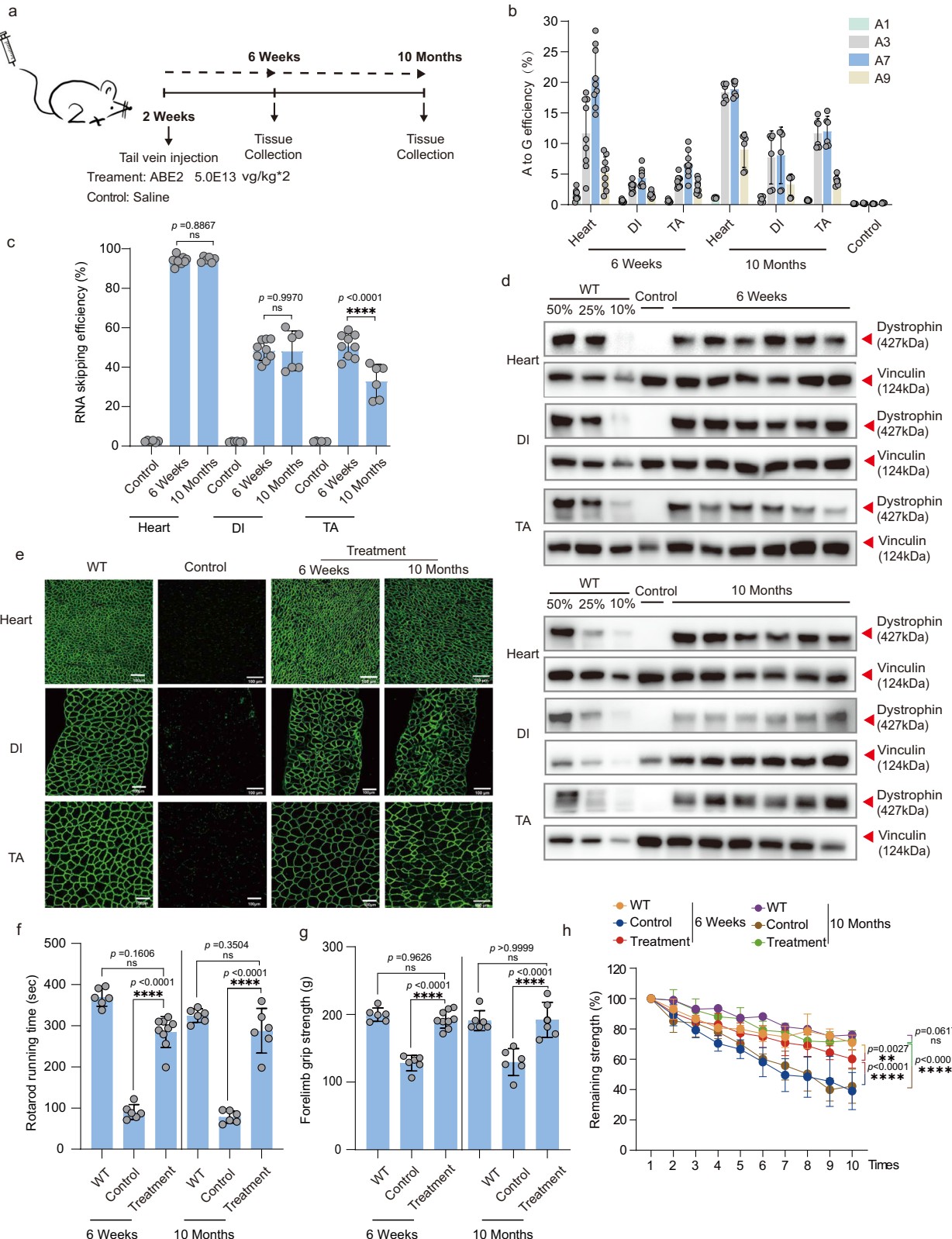

regaining dystrophin protein, leading to a higher number of Dys+ cells compared to the relatively modest DNA base editing efficiency[38].

Overall, our study provides proof-of-concept for treating the majority of *DMD* types caused by both exonic deletion and point mutations with human specific sgRNA guided ABE in genetically humanized DMD animals.

# Methods

## Ethical statement

The objectives of the present study were to generate a humanized DMD mouse model and obtain proof-of-concept demonstration of in vivo CRISPR-mediated base editing for DMD intervention. All animal experiments complied with the ARRIVE guidelines for reporting animal

**Fig. 5 | Intravenous systemic delivery of ABE system efficiently rescues dystrophin expression and muscle function in 2-week-old DMD mice. a** Schematic of intravenous administration of AAV-ABE2 particles. Tissues were collected for genomic DNA, RNA, immunoblotting and immunofluorescence experiments at 6-week and 10-month after treatment. Black arrows indicate time points for tissue collection after IV injection. **b** Measurement by deep sequencing of splicing site mutation efficiency in TA, DI, and heart after systemic delivery. RT-PCR products from muscle of DMD$^{\Delta mE50S1,KlhE50/Y}$ mice were analyzed by deep sequencing (**c**) to validate exon skipping efficiency. **d** Western blot analysis shows restoration of dystrophin expression in the tibialis anterior (TA), diaphragm (DI) and heart of DMD$^{\Delta mE50S1,KlhE50/Y}$ mice 6-week and 10-month after injection. Dilutions of protein extract from WT mice were used to standardize dystrophin expression (10%, 25%

and 50%). Vinculin was used as the loading control. **e** Immunohistochemistry for dystrophin in TA, DI, and heart of DMD$^{\Delta mE50S1,KlhE50/Y}$ mice was performed 6-week and 10-month after IP injection. Dystrophin is shown in green. Scale bar, 100 μm. Rotarod rod running time (**f**) and Forelimb grip strength (**g**) was measured two days in WT, DMD$^{\Delta mE50S1,KlhE50/Y}$ mice, and DMD$^{\Delta mE50S1,KlhE50/Y}$ mice treated with ABE2 particles. **h** The remaining strength was also measured during 10 repetitions at 10-s intervals. Each dot represents an individual mouse. Data are presented as mean ± s.d ($n = 6$ independent biological replicates for 10-month post-treatment group and $n = 9$ independent biological replicates for 6-week post-treatment group). Significance is indicated by an asterisk and determined using the one-way ANOVA multiple comparison test. *$P < 0.05$. **$P < 0.01$. ****$P < 0.0001$, Ns represents not statistically significant. Source data are provided as a Source Data file.

experiments. Duchenne muscular dystrophy (DMD) is the most common sex-linked lethal disease in human male patients, thus male mice were used in all experiments, such as grip tests, CK analysis and AAV injection. Specifically, we randomly assigned male DMD mice to either treatment or control groups to ensure an unbiased distribution. Furthermore, researchers responsible for assessing the outcomes were blinded to the group allocations. All animal experiments were performed in HuidaGene Therapeutics Inc., Shanghai, China and approved by our Institutional Animal Care and Use Committee (IACUC) under the approval number HGAF2021-008. Our animal facility was subjected to the annual review process conducted by the Shanghai Science and Technology Commission. The yearly evaluation covered various aspects including facility operations, animal care, and ethical reviews pertaining to the use of experimental animals. Only after passing this comprehensive inspection was the seal stamped, thereby extending the validity of our animal facility license for another year.

### Plasmid construction

The NG-ABE8e (Plasmid #138491) was a gift from David Liu lab, then a U6-driven crRNAs with a BpiI cloning site and CBh promoter driven EGFP were cloned into NG-ABE8e formed pU6-BpiI-CMV-ABE8e-CBh-EGFP vector. The sgRNAs targeting *DMD* gene exon2, exon45, exon50 and exon55 were designed using the CHOPCHOP tool, and synthesized as DNA oligonucleotides and cloned into pU6-BpiI-CMV-ABE8e-CBh-EGFP to form the CRISPR targeting plasmids (listed in Table S1). The N- and C-terminal intein sequences (Npu, Rma, Gry, Npu37, and Cfa) were custom-synthesized by Genewiz (Suzhou, China) and integrated into pU6-BpiI-CMV-ABE8e-CBh-EGFP backbones using Gibson cloning following PCR amplification. Additionally, highly efficient sgRNA targeting exon 50 of the *DMD* gene and muscle-specific promoter-driven Cas9-N or Cas9-C were generated in various AAV versions (ABE1 and ABE2) utilizing the Gibson assembly technique.

### Cell culture, transfection, and flow cytometry analysis

HEK293T cells sourced from the American Type Culture Collection (ATCC) were grown in Dulbecco's modified Eagle medium (DMEM) (Gibco, 11965092) containing 10% fetal bovine serum at a temperature of 37 °C in a humidified atmosphere with 5% CO2. CRISPR targeting different exons was transfected using polyethylenimine (PEI) transfection reagent (Polysciences, 24765-1). Following transfection, cells were cultured for 48 h, harvested, and analyzed or sorted using a BD FACSAria II flow cytometer. Cell lysis was carried out with specific primer sets (Table S1).

### Generation of DMD$^{\Delta mE50S1,KlhE50/Y}$ mice and animal experiments

Mice were housed in a controlled facility with a 12-h light/dark cycle and cared for in compliance with the Guidelines for Laboratory Animal Care set forth by the Ministry of Science and Technology of China. DMD$^{\Delta mE50S1,KlhE50/Y}$ mice were generated within the C57BL/6J strain using the CRISPR-Cas9 system, as described in our previous study[22,23]. Two sgRNAs were designed to target mouse *Dmd* intron 49 and intron 51 (Table S1), followed by addition of the T7 promoter sequence to the

sgRNA template. After purification of the PCR product with the Omega gel extraction kit (Omega, D2500-02), in vitro transcription was carried out using the MEGAshortscript T7 Kit (Invitrogen, AM1354). The sgRNAs were then purified using the MEGAclear Kit (Invitrogen, AM1908) and their concentration was determined using a NanoDrop instrument. Cytoplasmic injection involved a mixture of spCas9 mRNA (100 ng/μl), sgRNA (100 ng/μl), and HMEJ donor (100 ng/μl), which was injected into fertilized eggs and then transferred to recipient mice. Genomic DNA from the tail tissue of founder (F0) mice was isolated and analyzed (Table S1).

### AAV production and delivery to DMD$^{\Delta mE50S1,KlhE50/Y}$ mice

AAVs were produced with assistance from PackGene Biotech in Guangzhou, China. Transfection was performed at a confluency of 70–90%, and after three days, AAVs were purified using iodixanol density gradient centrifugation. For intramuscular injection, 3-week-old DMD$^{\Delta mE50S1,KlhE50/Y}$ mice were anesthetized and treated with AAV preparations in the TA muscle or saline solution. Systemic injections were administered to DMD$^{\Delta mE50S1,KlhE50/Y}$ mice with 1.0E14 vg/kg viral particles via via tail vein (2 weeks old mice) and intraperitoneal approach (P4 mice). Tissue samples were collected for various experiments at predetermined time points post-treatment.

### Satellite cell isolation and culture

The hindlimb and forelimb skeletal muscles were removed after the mice were euthanized. The muscles were washed twice with Dulbecco's phosphate-buffered saline (DPBS) (Thermo Fisher Scientific, 14190144) and finely chopped. Subsequently, they were digested using a 0.2% Collagenase type 2 (Gibco, 17101015) solution for 60 min in a shaking water bath at 37 °C. A second digestion was performed with a solution consisting of 0.2% Collagenase type 2 and 0.4% Dispase (Gibco, 17105041) in Rinsing media for 30 min in a shaking water bath at 37 °C. The digested tissue was then passed through a 40 mm filter to collect the filtrate. In order to increase the probability of available satellite cells, a purification process was conducted by wall sticking screening twice, each time for 1 h. The prepared cell suspension was then inoculated into ECM-coated (Sigma, E0282) coverslips. The satellite cells were cultured in growth medium (Ham's F10, 10% fetal bovine serum) supplemented with fibroblast growth factor (FGF) (Gibco, 13256-029) at 37 °C in a humidified incubator with 5% CO2.

The immunofluorescence detection method for satellite cells was as follows. Cells were fixed with 4% paraformaldehyde at room temperature (RT) for 15 min, followed by three washes with 1× PBS, each lasting 5 min. The cells were then blocked using an immunostaining blocking solution for 60 min at RT. Subsequently, the cells were incubated overnight at 4 °C with primary antibodies against PAX7 (DSHB, 042349, 1:500) and PAX3 (Beyotime, AF7686, 1:500) diluted appropriately in PBS. After three rinses in PBS, each for 5 min, the cells were stained with a fluorochrome-conjugated secondary antibody, diluted appropriately in PBS, for 1 h at 37 °C in the dark. Finally, the specimens were incubated with 1 μg/ml DAPI for 10 min to stain the nuclei, and were immediately examined using fluorescence microscopy.

## Targeted deep sequencing

To assess the efficacy of A-to-G base editing, the genetic material from successfully transfected cells or tissues treated with ABE1 or ABE2 was extracted following the recommended guidelines provided by the manufacturer. Subsequently, the DNA samples were subjected to amplification using specific primer sets, and the resulting products were analyzed using deep sequencing techniques. In order to evaluate the efficiency of exon skipping, total mRNA was isolated from muscle tissue and converted to cDNA utilizing the HiScript II One Step RT-PCR Kit (Vazyme, P611-01), in accordance with the manufacturer's instructions. The synthesized cDNA was then amplified using Phanta Max super-fidelity DNA polymerase (Vazyme, P505-d1) for subsequent analysis through Sanger sequencing, gel electrophoresis, and deep sequencing methodologies. The deep sequencing libraries were prepared by incorporating Illumina flow cell binding sequences and specific barcodes at the 5′ and 3′ ends of the primer sequences. After pooling the samples, sequencing was performed using 150 paired-end reads on an Illumina HiSeq platform. The raw data in FASTQ format were processed using Cutadapt (v.2.8)[39] and analyzed with CRISPResso2[40], based on the assigned barcode sequences.

## Off-target analysis

For off-target analysis, the PAM sequence (5′-NG-3′) of XCas9 3.7 (TLIKDIV SpCas9) sourced from *Streptococcus pyogenes* was utilized for predictive purposes using the Cas-OFFinder tool (http://www.rgenome.net/cas-offinder/). Subsequently, 14 potential off-target sites with 3 mismatched positions were selected for further scrutiny. Targeted amplification of all predicted off-target loci containing no more than 3 mismatches in relation to the target site was carried out for subsequent deep sequencing analysis.

## Western blot analysis

Tissue samples were processed using RIPA buffer supplemented with a protease inhibitor cocktail to ensure efficient homogenization. The resulting lysate supernatants underwent quantification utilizing a Pierce BCA protein assay kit (Thermo Fisher Scientific, 23225) and were subsequently adjusted to a consistent concentration with deionized water. Following this, equivalent quantities of the samples were combined with NuPAGE LDS sample buffer (Invitrogen, NP0007) and 10% β-mercaptoethanol, leading to a boiling step at 70 °C for a duration of 10 min. Subsequently, 10 μg of total protein per lane was loaded into 3% to 8% tris-acetate gels (Invitrogen, EA03752BOX) and subjected to electrophoresis at 200 V for a period of 1 h. The proteins were then transferred onto a PVDF membrane in a wet transfer system at 350 mA for 3.5 h. The membrane was subjected to blocking in 5% non-fat milk dissolved in TBST buffer, followed by incubation with a primary antibody targeting the specific protein of interest. Subsequent to washes with TBST, the membrane underwent incubation with an HRP-conjugated secondary antibody that was specific to the IgG of the species corresponding to the primary antibody against dystrophin (Sigma, D8168) or vinculin (CST, 13901 S). Detection of the target proteins was achieved using chemiluminescent substrates (Invitrogen, WP20005).

The quantitative analysis of Western blotting bands was conducted as follows: the intensities of the dystrophin bands for varying percentages (50%, 25%, 10%, or 75%) of WT muscle protein lysate, as well as Control-, ABE1-, and ABE2-treated muscle lysates, were quantified using ImageJ software. The dystrophin band intensity was normalized to that of Vinculin as the internal loading control. Subsequently, the dystrophin levels in the control or treated DMD mice were further normalized to those in WT mice. The outcomes were expressed as a percentage of the WT dystrophin level for each specific lane.

## Histology and Immunofluorescence

Tissues were harvested and immersed in preconditioned 4% paraformaldehyde for fixation. Post-fixation, the tissues underwent a graded alcohol dehydration process from low to high concentrations. Following xylene treatment, the specimens were embedded in paraffin wax, sectioned into 10 μm slices, and mounted onto slides. Paraffin was subsequently removed using xylene, and slides were rehydrated through alcohols of decreasing concentration, concluding with distilled water. For hematoxylin and eosin (H&E) staining, slides were stained with hematoxylin for 3–8 min, followed by differentiation using acid and ammonia water. The slides were then dehydrated sequentially in 70% and 90% alcohol for 10 min each, stained with eosin for 1–3 min, and further dehydrated in graded alcohols (50%, 70%, 80%, 95%, and 100%). Coverslips were applied with neutral resin. To quantify centrally nucleated fibers (CNFs), H&E-stained slides were examined under a light microscope at 20× magnification. Each muscle fiber's nucleus positioning was assessed, with a threshold of 500 fibers per sample ensuring statistical accuracy. CNFs were defined by nuclei located within the central one-third of the fiber's cross-sectional area, and their percentage was calculated by dividing the number of CNFs by the total fiber count, multiplied by 100. For Sirius red staining, slides were immersed in picrosirius red for one hour, followed by rinsing in acidified water. Excess water was removed, and slides were dehydrated in three changes of absolute ethanol, cleared in xylene, and mounted with neutral resin.

For immunofluorescence, tissues were embedded in optimal cutting temperature (OCT) compound and flash-frozen in liquid nitrogen. Cryosections (10 μm) were optionally fixed for 2 h at 37 °C, permeabilized with PBS containing 0.4% Triton-X for 30 min, and blocked with 10% goat serum for one hour at room temperature. The slides were incubated overnight at 4 °C with primary antibodies against dystrophin (Abcam, ab15277, 1:100) and spectrin (Millipore, MAB1622, 1:500). The following day, samples were washed with PBS, incubated with Alexa Fluor® 488 donkey anti-rabbit IgG (Jackson ImmunoResearch labs, 711-545-152, 1:500) or Alexa Fluor 647 donkey anti-mouse IgG (Jackson ImmunoResearch labs, 715-605-151, 1:500) and DAPI for 3 h at room temperature. After extensive PBS washing, slides were sealed using fluoromount-G mounting medium and visualized using an Olympus FV3000 or Nikon C2 microscope. The percentage of Dys+ muscle fibers was calculated relative to total spectrin-positive muscle fibers.

## Forelimb grip strength and rotarod test

Muscle strength was evaluated at multiple intervals. In brief, mice were extracted from their enclosures, weighed, and held by the tail, prompting their forelimbs to grip the pull-bar apparatus linked to the 47200-grip strength meter. The mice were then drawn along a straight trajectory away from the sensor until they could no longer maintain their grip, at which point the peak force exerted, measured in grams, was recorded. This procedure was repeated 7 to 10 times with 10-s rest periods between trials.

For the rotarod test, mice underwent daily 30-min training sessions during the week preceding the experiment. During the actual test, the mice were placed on the rotarod (Ugo Basile Inc.), which accelerated from 4 to 40 rpm over a span of 30 s. Four mice were tested concurrently, and each mouse underwent five trials. Latency was defined as the duration from the onset of the trial (initiation of rod acceleration) until the mouse fell off and triggered the lever that stopped the timer.

## Serum ALT, BNU and CK analysis

To assess CK levels, a blood sample was procured via cardiac puncture under ketamine anesthesia before euthanasia, using an Eppendorf tube. The collected blood samples were then centrifuged at 3,000 × g for 10 min, allowing for serum separation. Serum CK activity was quantified employing creatine kinase reagent (Pointe Scientific, 23-666-208), adhering strictly to the manufacturer's protocol. The detection process was executed using the Chemray 800, a fully automated biochemical analyzer.

## Statistics and reproducibility

The data are presented as mean ± s.d. The number of independent biological replicates are shown in figure legends. No data were excluded from the analyses. We randomly selected cells for test group and control group. DMD mice used for gene editing therapy were allocated to control or AAV9-treated group randomly. Differences were assessed using unpaired two-tailed Student's $t$ test or one-way ANOVA. Differences in means were considered statistically significant when they reached $P < 0.05$. Significance levels are $*P < 0.05$. $**P < 0.01$. $***P < 0.001$. $****P < 0.0001$.

## Reporting summary

Further information on research design is available in the Nature Portfolio Reporting Summary linked to this article.

## Data availability

All the sequencing data have been deposited in the NCBI SRA repository under project accession number PRJNA1105327. Source data are provided with this paper.

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

## Acknowledgements

This work was supported by the Lingang Laboratory fund (LG202106-01-02) (H.Y.), National Science and Technology Innovation 2030 Major Program (2021ZD0200900) (H.Y.), Chinese National Science and Technology major project R&D Program of China (2018YFC2000101) (H.Y.), Strategic Priority Research Program of Chinese Academy of Science (XDB32060000) (H.Y.), National Natural Science Foundation of China (31871502, 31901047, 31925016, 91957122, and 82021001) (H.Y.), (82202052) (M.J.) and (81870902) (N.W.), Basic Frontier Scientific Research Program of Chinese Academy of Sciences From 0 to 1 original innovation project (ZDBS-LY-SM001) (H.Y.), Shanghai Municipal Science and Technology Major Project (2018SHZDZX05) (H.Y.), Shanghai City Committee of Science and Technology Project (20ZR1466600, 18411953700, 18JC1410100, 19XD1424400, and 19YF1455100) (H.Y.) and the International Partnership Program of Chinese Academy of Sciences (153D31KYSB20170059) (H.Y.). Joint Funds for the Local Science and Technology Development Project guided by the central government grants 2022L3011 (N.W.). And the Youth Scientific Research Project of Fujian Provincial Health Commission (2020QNA045) (M.J.). C.X. is sponsored by the Lingang Laboratory Intramural Fund and Project of Shanghai Municipal Science and Technology Commission (22QA1412300).

## Author contributions

H.Y., G.L. and C.X. jointly conceived the project and designed experiments. H.Y., C.X., W.C. and N.W. supervised the project. G.L. and Z.L. generated the humanized mouse model. G.L., D.Y., J.L. and Y.Z. designed vectors, performed in vitro experiments and conducted confocal imaging. C.X. assisted with the construction of plasmids. G.L. and J.L. performed in vivo virus injection, tissue dissection, histological immunostaining and muscle function experiments. Q.X., J.L., Y.W., Y.Y., X.Z. and D.Y. assisted with tissue dissection, immunostaining, and animal breeding. G.L., M.J., J.L., D.Y. and C.X. analyzed the data and organized figures. H.Y., C.X. and G.L. wrote the manuscript with data contributed by all authors who participated in the project.

## Competing interests

H.Y. is a founder of HuidaGene Therapeutics. The remaining authors declare no competing interests.
