## [Peer Review File · Nature Communications]

Reviewers' Comments:

Reviewer #1:

Remarks to the Author:

The authors describe an adenine base editor-mediated exon skipping approach targeting the human DMD exon 50 splice donor site. Using a plasmid-based system, they first screen for suitable sgRNAs and intein-split ABEs in FACS-sorted HEK293T in vitro. In the next step, the authors test their genome editing strategy in a newly established and characterized humanized mouse model carrying the human DMD exon 50 sequence alongside an exon 51 knock-out. Skipping of exon 50 would restore an open reading frame in a shortened variant of Dmd, and thus might present a strategy to improve Dmd protein levels and ultimately the muscle phenotype.

The authors demonstrate A to G conversion at the on-target site, efficient exon 50 skipping at the mRNA-level, and rescue of Dmd expression in Western Blots and IF stainings after AAV-delivery i.m., i.p., or i.v.. The authors evaluate several muscle groups and perform functional muscle strength and mobility tests that show an improvement over untreated and saline mock-treated controls

Has the new mouse model been characterized before or after the age of 8 weeks (the age where analyses were performed, legend Fig. 1). Please comment on the reasons why this time point was chosen. The presence and abundance of revertant fibers should be commented on.

Histology Fig. 1: Severe fibrosis is not shown in the TA.

Fig. 1H: The mean and SEM of CK levels do not reflect the data points shown.

Discuss differences between human and murine exon 50. In which aspect is the model humanized?

Fig. 3C: Please comment on the additional band in the control-panel. The band demonstrates that you see exon 51 KO and in addition exons 50 plus 51 KO.

Fig. 3B shows an A>G conversion of 20% for ABE1 and 25% for ABE2. How do the authors then explain 80 – 100% RNA skipping efficiency? Are there bystander edits? Please provide a chromatogram of the edited and unedited splice donors sites.

Fig. 4B and D: Please show bars on the same scale from 1 to 100% (same for 5B and D)

Fig. 4C and 5C: The sizes of the PCR products are not comprehensible. Please list all primers in the Supplement or name, if a different marker was used. A scheme of the size of the expected PCR products would ease the reading of the manuscript.

Fig. S3: Please describe how the off-target analysis was performed. How many mismatches were allowed? How were the eight sites selected? Which PAM-specification was used for the prediction. The (Cas)-OFFinder is not cited (Bae et al, 2014).

There is no mentioning of skipping in the muscle stem cells, satellite cells.

How long does the effect of skipping last? Was six weeks after editing the only timepoint that was checked? Please clarify how i.v. delivery was performed: retro-orbital or via tail vein?

The claims in the discussion should be down-sized. It has not been examined whether gene editing byproducts were produced. Also, whether this is a safe alternative to existing strategies is also questionable. The topic of off-target analysis is discussed only very briefly. The specific ABE that was used is not discussed. The very similar paper of Chemello et al. 2021; DOI: 10.1126/sciadv.abg4910 has not been cited. The major difference to the paper by Chemello et al. is the fact that there was no humanized exon 50 in the Chemello-paper.

Minor:

Remove Company names from the legend

Fig. 4F: Please add a legend to the columns either in the figure itself or in the legend text.

Fig. S6: The legend does not correspond to the figure.

Reviewer #2:

Remarks to the Author:

Reviewer #3:

Remarks to the Author:

This paper presented the generation of a humanized mice model of DMD by replacing mouse dmd exons 50 and 51 with human DMD exon 50, and the correction of this model by ABE-mediated exon 50 skipping. The authors demonstrated that this humanized mouse model recapitulated DMD patients' phenotypes and muscle dysfunction. The authors also demonstrated that systemic delivery of dual AAVs expressing ABE and guide RNA targeting exon 50 splicing sites could rescue the phenotypes and restore muscle function. Overall, the manuscript presents valuable research on DMD gene correction and addresss an important scientific qesiton.

However, this reviewer have concerns regarding the novelty of this study. It is crucial to address these concerns to ensure the credibility and reliability of the publication. The following comments aim to assist the authors in improving the manuscript in this regard:

Major comments:

Novelty:

The manuscript lacks a clear demonstration of how it significantly advances the field or offers novel insights. Several other groups have published CRISPR-mediated approaches to generate and correct humanized dystrophic animal models of DMD, as summarized in a review article (PMCID: PMC7141101, table 2). Additionally, other papers have been published on the application of ABE/CBE to modify DMD splice sites in mice (PMCID: PMC8087404) and human cardiomyocytes (PMCID: PMC9792405). It would be beneficial if the authors could cite these similar works in this manuscript to provide a broader context and demonstrate the novelty of their study.

Methodological Transparency:

The manuscript would benefit from providing additional details to ensure methodological transparency.

a) In Figure 5A, the authors show TAIL VEIN INJECTION for two-week-old mice with 5×10^{13} vg/kg*2. However, the Methods and Materials section (line 652-654) states that the DMD Δ mE5051, KIhE50/Y mice were injected systemically via a retro-orbital approach (2 weeks old). While the retro-orbital injection method was not mentioned in the main text and figures, and the details for tail vein injection for two-week-old mice are missing in the method section. Clarifying these details is essential to ensure reproducibility and clarity for other researchers attempting to repeat these experiments.

b) In line 145, the authors state that ABE1 only contains one copy of sgRNA, while ABE2 has two

sgRNAs. However, in Figure 2D, ABE1 contains two copies of sgRNA in Cas9n-N, and ABE2 contains three copies of sgRNA. This inconsistency between the main text, figure legends, and the figure itself could cause confusion for readers and researchers trying to understand the methodology. It is crucial to correct this inconsistency and provide accurate information on copy numbers for sgRNAs used in each AAV construct.

c) To support the results, the manuscript should include a clear method for quantifying Western blot bands. For example, in Figure 3H, the dystrophin expression of the ABE2 group appears higher than 50% of the wild type (WT), even after normalization to Vinculin expression. However, Figure 3I shows that ABE2 only has approximately 40% dystrophin expression compared to WT. Including a description of the Western blot band quantification method would enhance the transparency and reliability of the results.

Minor comments:

In Figure 1F, the muscle tissue type should be labeled to provide clarity.

All Western gels and PCR gels should include molecular weight markers for accurate size estimation. Full blot image of Western blot should be presented in Suppl.

Figure 4F is missing the mice type label, which should be added for clarity.

In Figure 5C, "teatement" is labeled on molecular markers. Please fix it.

The figure legends in supplemental Figure S6 and S9 do not match the figure panels. Please ensure that the figure legends accurately describe the content of the figures.

Other suggestion (optional)

CK levels should be measured in the systemic delivery mice, but this reviewer understand that the blood samples might not be available anymore since the experiments already finished.

Reviewer #4:

Remarks to the Author:

This manuscript reports a novel humanized mouse model of DMD and an adenine base editor (ABE) strategy to induce exon skipping. They show a remarkable reconstitution of dystrophin expression depending on the construct design.

Some limitations should be addressed:

1. Overall, there are many gene editing approaches in DMD models. What is the particular advantage and novelty compared to e.g. PMID: 33931459?
2. The methods section states in line 592 that "All animal experiments were performed and approved by the Institutional Animal Care and Use Committee (IACUC) of HuidaGene Therapeutics Inc., Shanghai, China." I assume that this statement is not correct as the IACUC probably did not perform the animal experiments. Furthermore, it is unusual that a company board approves animal experiments and not governmental or university bodies. In addition, no reference to international guidelines of animal experiments are made.
3. Animal numbers are very small (n=3) raising statistical concerns. In addition, display of SEM instead of standard deviations is not correct.
4. Information regarding the cardiac phenotype is lacking, e.g. left ventricular function. Have authors characterized the heart in their model at baseline and in the gene therapy studies?
5. How do you explain the striking difference in efficiency between ABE1 and ABE2?

REVIEWER COMMENTS

Reviewer #1 (Remarks to the Author):

The authors describe an adenine base editor-mediated exon skipping approach targeting the human DMD exon 50 splice donor site. Using a plasmid-based system, they first screen for suitable sgRNAs and intein-split ABEs in FACS-sorted HEK293T in vitro. In the next step, the authors test their genome editing strategy in a newly established and characterized humanized mouse model carrying the human DMD exon 50 sequence alongside an exon 51 knock-out. Skipping of exon 50 would restore an open reading frame in a shortened variant of Dmd, and thus might present a strategy to improve Dmd protein levels and ultimately the muscle phenotype.

The authors demonstrate A to G conversion at the on-target site, efficient exon 50 skipping at the mRNA-level, and rescue of Dmd expression in Western Blots and IF stainings after AAV-delivery i.m., i.p., or i.v.. The authors evaluate several muscle groups and perform functional muscle strength and mobility tests that show an improvement over untreated and saline mock-treated controls

Response: Thanks for the positive comments and the relevant advice.

Has the new mouse model been characterized before or after the age of 8 weeks (the age where analyses were performed, legend Fig. 1). Please comment on the reasons why this time point was chosen. The presence and abundance of revertant fibers should be commented on.

Response: Thank you for the good comment. We have performed phenotypic characterization of the humanized mouse model at 2-week, 8-week and 24-week. We only showed the results of 8-week DMD mice in Fig.1. In the revised manuscript, we have provided all results shown as follow (**Fig. R1**) for comprehensive phenotypic characterization of DMD mice at different time points. Briefly, the muscular histology of the DMD mice showed the inward migration of cell nuclei in all the tested muscle tissues occurred as early as 2 weeks old (**Fig. R1a**). As the mice grew older (8 weeks and 24 weeks old), the populations of centrally nucleated fibers (CNFs) significantly increased and the inflammatory cell infiltration also appeared at 2 weeks of age, with the increasing severity over time (**Fig. R1b**). The appearance of creatine kinase (CK) in blood has been considered as a biochemical marker of muscle necrosis. Consistent with dystrophin deficient DMD patients, serum CK activity of this humanized DMD mice had a dramatic elevation at 2 weeks of age when compared to the wildtype (WT) controls, indicating the severity of muscle damage (**Fig. R1c**). Moreover, the DMD mice exhibited significant reduction in muscle strength as early as 6 weeks old when compared to the age-matched WT mice (**Fig. R1d**), recapitulating progressive muscle weakness of DMD patients. Therefore, 8-week-old DMD mice were chosen for evaluation. In the humanized DMD mice, we detect very few revertant fibers that are difficult to be distinguished from background antibody staining signal (**Fig. R2**).

Fig. R1: The muscle pathology and motor function of DMD mice at different ages. **a**, Hematoxylin and Eosin staining of diaphragm (DI), gastrocnemius (GA), and quadriceps (QA), and heart muscle of WT and different age of DMD mice. WT mice as control. Scale bar, 200 μ m. $n = 3$. Yellow arrowhead, inward migration of nuclei; Green arrowhead, inflammatory cell infiltration. **b**, Statistical analysis of nuclear migration in H&E staining. **c**, Serum CK, a marker of muscle damage and membrane leakage, was measured in WT and DMD mice at the ages from 2 weeks to 24 weeks. $n = 6$. **d**, The forelimb grip strength testing to measure muscle performance of WT and DMD mice at the ages from 2 weeks to 24 weeks. $n = 6$. Data are presented as mean \pm SEM. Each dot represents an individual mouse. Significance is indicated by asterisk and determined using unpaired two-tailed Student's t test. Scale bar, 200 μ m.

Fig. R2: Dystrophin staining results in heart, TA and DI from wildtype and DMD mice of different ages.

Histology Fig. 1: Severe fibrosis is not shown in the TA.

Response: Thanks for your comment. Through Sirius red and HE staining, we observed severe fibrosis in the diaphragm and anterior tibialis muscles. The H&E and Sirius red staining results of 8 DMD mice are presented as follow (**Fig. R3**), and representative results have been selected to replace Fig.1e.

Fig. R3: The H&E and Sirius red staining results of wildtype and DMD mice.

Fig. 1H: The mean and SEM of CK levels do not reflect the data points shown.

Response: Thanks for raising this issue. We have replotted the results (**Fig. R4**) as suggested to revise the mistake.

Fig. R4: Serum CK measured in WT and DMD mice.

Discuss differences between human and murine exon 50. In which aspect is the model humanized?

Response: Although the homology between mouse and human DMD genes is very high, there are still significant sequence differences near the splicing site (**Fig. R5**). To generate a genetically humanized DMD mouse model carrying specific human exon deletion

mutations, we knocked in human exon 50 with flanking 200 bp sequences to replace mouse exons 50 and 51 in a single step. This will facilitate the study of gene-editing-mediated exon skipping therapy.

Fig. R5: Alignment of human and mouse exon 50 sequence. Human exon 50 (Yellow labeling), mouse exon 50 (Blue labeling), sgRNA (magenta line), Protospacer adjacent motif (Red box).

Fig. 3C: Please comment on the additional band in the control-panel. The band demonstrates that you see exon 51 KO and in addition exons 50 plus 51 KO.

Response: Thanks for raising this issue. We sequenced the unexpectedly appearing band and found that this sequence was generated by exon skipping. We speculate that it is generated by spontaneous exon skipping in mice.

Fig. 3B shows an A>G conversion of 20% for ABE1 and 25% for ABE2. How do the authors then explain 80 – 100% RNA skipping efficiency? Are there bystander edits? Please provide a chromatogram of the edited and unedited splice donors sites.

Response: Thanks for your good comment. We also find the discrepancy between genome editing and RNA skipping efficiency intriguing. Previous study (PMID: 34698513 , PMID: 30854433) reported similar results with us, which might be due to different stability of edited and unedited transcripts or methodologic difference between DNA and RNA editing analysis. For bystander editing analysis, we provided the reads analysis presented as follow (**Fig. R6**) to show alleles with or without bystander editing events.

Fig. R6: Deep-seq reads analysis of ABE1- and ABE2-edited DMD gene in DMD mice.

Fig. 4B and D: Please show bars on the same scale from 1 to 100% (same for 5B and D)

Response: Thanks for the suggestion. The Fig. 4B and D, Fig. 5B and D have been modified as suggested (Fig. R7).

Fig. R7: Genome editing and RNA skipping efficiency after intraperitoneal administration of AAV-ABE2. a, Genome editing efficiency heatmap in heart, DI and TA for control and treated mice. **b,** Percentage of RNA skipping in heart, DI and TA for control and treated mice. n = 3. Data are represented as mean \pm SD. Each dot represents an individual mouse.

Fig. 4C and 5C: The sizes of the PCR products are not comprehensible. Please list all primers in the Supplement or name, if a different marker was used. A scheme of the size of the expected PCR products would ease the reading of the manuscript.

Response: Thank you for your careful review and helpful suggestions. The sizes of the PCR products were added in Fig. 4C and 5C shown as follow (**Fig. R8**), and all primers were also provided in Supplementary Table S1 of the revised manuscript.

Fig. R8: RT-PCR products from muscle of DMD mice were analyzed by gel electrophoresis. 449 bp and 340 bp bands are from transcripts without and with exon 50 skipping respectively.

Fig. S3: Please describe how the off-target analysis was performed. How many mismatches were allowed? How were the eight sites selected? Which PAM-specification was used for the prediction. The (Cas)-OFFinder is not cited (Bae et al, 2014).

Response: Thank you for your comment. We have revised the method description for off-target analysis. Using the PAM sequence (5'-NG-3') of XCas9 3.7 (TLIKDIV SpCas9) from *Streptococcus pyogenes* for prediction on the Cas-offinder website, all 14 potential off-target sites with 3 mismatched positions are selected for evaluation. In the revised manuscript, we have included all 14 predicted off-target sites with 3 mismatches and analyzed them using deep sequencing, showing high on-target editing with sgRNA6 but undetectable off-target editing events as follow (**Fig. R9**). In addition, we have cited the relevant paper as suggested.

Fig. R9: Off-target analysis of ABE2-mediated base editing for DMD gene. **a**, Alignment of the top 14 off-target sites in human genome. The target adenine (A7) is colored red. **b**, Percentages of adenine editing in the all 14 off-target sites. Dots and bars represent biological replicates and data are presented as means \pm SD (n = 3).

There is no mentioning of skipping in the muscle stem cells, satellite cells.

Response: Thanks for raising this issue. Dystrophin is expressed in differentiated myofibers and activated muscle stem cells (PMCID: PMC4839960). After intravenous injection of DMD mice with AAV-ABE2 for 6 weeks, we dissected the tibialis anterior muscle and isolated satellite cells for deep sequencing and RT-PCR analysis. The results showed that satellite cells have comparable A-to-G editing efficiency of up to 30% as the tibialis anterior muscle (**Fig. R10**). However, there is a low amount of mouse satellite cells, making it difficult to extract RNA and perform effective gel electrophoresis analysis of RNA skipping rate. Previous studies have reported that by using adeno-associated virus (AAV) to deliver gene editing tools targeting the DMD mutation region, it can be effectively delivered to muscle stem cells, allowing for gene editing and restoring their normal differentiation and renewal functions (PMID: 36995603, PMC4924477). Indeed, we think that your suggestion is very important and have taken it seriously in our lab to optimize satellite cells protocol for RNA skipping evaluation and hopefully get it done in another study.

Fig. R10: Gene editing analysis in satellite cells of DMD mice. **a**, Satellite cells of DMD mice treated with AAV-ABE2 showed PAX7 expression. **b**, Gene editing efficiency measured with deep-seq for satellite cells of DMD mice treated with AAV-ABE2. **c**, Deep-seq reads analysis for ABE2-edited DMD gene in satellite cells.

How long does the effect of skipping last? Was six weeks after editing the only timepoint that was checked? Please clarify how i.v. delivery was performed: retro-orbital or via tail vein?

Response: To examine the long-term therapeutic effect of ABE2 treatment, DMD mice with or without ABE2 administration were monitored for 10 months and then euthanized to analyze muscle tissues. Our results revealed durable therapeutic efficacy of ABE2 administration for 10 months, which are added in the revised manuscript presented as follow (Fig. R11).

Fig. R11: Intravenous delivery of ABE system efficiency rescues dystrophin expression and muscle function in humanized DMD mice. **a**, Schematic of intravenous administration of ABE2 particles. Tissues were collected for genomic DNA, RNA, immunoblotting and immunofluorescence experiments at 6 weeks (n=3) and 10 months (n=6) after treatment. Black arrows indicate time points for tissue collection after IV injection. **b**, Measurement by deep sequencing of splicing site editing efficiency in TA, DI, and heart after systemic delivery of ABE2. RT-PCR products from muscle of DMD^{ΔmE5051,K1hE50/Y} mice were analyzed by deep sequencing. **c**, RNA exon-skipping

efficiency. **d**, Immunohistochemistry for dystrophin in TA, DI, and heart of $DMD^{\Delta E5051, K1hE50/Y}$ mice was performed 6 weeks or 10 months after intravenous injection. Dystrophin is shown in green. Scale bar, 200 μ m. **e-f**, Western blot analysis shows restoration of dystrophin expression in the TA, DI, and heart of $DMD^{\Delta E5051, K1hE50/Y}$ mice 6 weeks or 10 months after injection. Dilutions of protein extract from WT mice were used to standardize dystrophin expression (10%, 25%, and 50%). Vinculin was used as the loading control. Forelimb grip strength (**g**) and rotarod rod performance (**h**) were measured two days in WT, and $DMD^{\Delta E5051, K1hE50/Y}$ mice treated without or with ABE2 particles. **i**, The remaining strength was measured during 10 repetitions at 10-second intervals. Dots and bars represent biological replicates and data are presented mean \pm SD. Significance is indicated by asterisks and was determined in Fig. 5g, h using unpaired two-tailed Student's t test or in Fig. 5i using ANOVA multiple comparison test.

The claims in the discussion should be down-sized. It has not been examined whether gene editing byproducts were produced. Also, whether this is a safe alternative to existing strategies is also questionable. The topic of off-target analysis is discussed only very briefly. The specific ABE that was used is not discussed. The very similar paper of Chemello et al. 2021; DOI: 10.1126/sciadv.abg4910 has not been cited. The major difference to the paper by Chemello et al. is the fact that there was no humanized exon 50 in the Chemello-paper.

Response: Thanks for helping improve our study. We have revised our manuscript as suggested and cited relevant papers.

Minor:

Remove Company names from the legend

Response: Thank you for your careful review and helpful suggestions. Company names in the legend were removed as suggested.

Fig. 4F: Please add a legend to the columns either in the figure itself or in the legend text.

Response: The legend text has been added to the figures of our revised manuscript.

Fig. R12: Immunohistochemistry for dystrophin in TA, DI, and heart of DMD mice was performed 6 weeks after IP injection. Dystrophin is shown in green. Scale bar, 200 μ m.

Fig. S6: The legend does not correspond to the figure.

Response: We felt sorry for the mistake. The manuscript was revised as commented.

Reviewer #2 (Remarks to the Author):

Reviewer #3 (Remarks to the Author):

This paper presented the generation of a humanized mice model of DMD by replacing mouse *dmd* exons 50 and 51 with human DMD exon 50, and the correction of this model by ABE-mediated exon 50 skipping. The authors demonstrated that this humanized mouse model recapitulated DMD patients' phenotypes and muscle dysfunction. The authors also demonstrated that systemic delivery of dual AAVs expressing ABE and guide RNA targeting exon 50 splicing sites could rescue the phenotypes and restore muscle function. Overall, the manuscript presents valuable research on DMD gene correction and address an important scientific question.

Response: Thanks for the positive comments and the relevant advice.

However, this reviewer have concerns regarding the novelty of this study. It is crucial to address these concerns to ensure the credibility and reliability of the publication. The following comments aim to assist the authors in improving the manuscript in this regard:

Major comments:

Novelty:

The manuscript lacks a clear demonstration of how it significantly advances the field or offers novel insights. Several other groups have published CRISPR-mediated approaches to generate and correct humanized dystrophic animal models of DMD, as summarized in a review article (PMCID: PMC7141101, table 2). Additionally, other papers have been published on the application of ABE/CBE to modify DMD splice sites in mice (PMCID: PMC8087404) and human cardiomyocytes (PMCID: PMC9792405). It would be beneficial if the authors could cite these similar works in this manuscript to provide a broader context and demonstrate the novelty of their study.

Response: Thanks for raising this issue. We have cited the relevant study as suggested in the revised manuscript.

Overall, our study introduces several innovative aspects in the field of Duchenne Muscular Dystrophy (DMD) research. Firstly, it recognizes the significant differences between human and mouse DMD gene sequences. The generation of a humanized DMD mouse model is critical for evaluating the preclinical efficacy of drugs within mice. We successfully created a humanized DMD model by one-step knockout of mouse DMD exons 50 and 51, and insertion of human exon 50. This model exhibited pathological characteristics consistent with DMD patients.

Secondly, we efficiently achieved DMD exon 50 skipping through the optimization of the AAV-ABE vector and inteins split strategy. This was effective regardless of whether it was delivered via intramuscular injection or systemic delivery.

Thirdly, while previous studies have reported the application of ABE/CBE to modify DMD splice sites in mice (PMCID: PMC8087404/PMID: 33931459) and human cardiomyocytes (PMCID: PMC9792405), they only implemented it through intramuscular injection of TA muscle or in vitro assay, but did not evaluate the recovery of motor function and serum biomarkers via systemic delivery. In contrast, our manuscript systematically evaluates the therapeutic effects of the gene editing tool through both intraperitoneal and tail vein injections in the DMD model, and tracks the exon 50 skipping up to 10 months, demonstrating the long-term effectiveness of strategy via base-editing mediated exon-skipping.

These innovative aspects make this manuscript highly significant in the field of DMD research. It not only increases our understanding of the differences between human and mouse DMD gene sequences but also provides a reliable humanized DMD mouse model for more accurate evaluation of drug development. Additionally, the efficient exon 50 skipping achieved through AAV-ABE optimization offers a new approach for gene editing therapy in DMD. Most importantly, the manuscript extensively investigates the long-term effectiveness of base-editing, providing strong support for future clinical applications.

Methodological Transparency:

The manuscript would benefit from providing additional details to ensure methodological transparency.

Response: Thanks for the suggestion. We have rewritten the sections on injection methods and analysis techniques in the revised manuscript.

a) In Figure 5A, the authors show TAIL VEIN INJECTION for two-week-old mice with 5×10^{13} vg/kg*2. However, the Methods and Materials section (line 652-654) states that the DMD Δ mE5051,KIhE50/Y mice were injected systemically via a retro-orbital approach (2 weeks old). While the retro-orbital injection method was not mentioned in the main text and figures, and the details for tail vein injection for two-week-old mice are missing in the method section. Clarifying these details is essential to ensure reproducibility and clarity for other researchers attempting to repeat these experiments.

Response: Thanks for the suggestion. The manuscript did not utilize retro-orbital injection method. Systemic injections mainly included intraperitoneal injection in 3-day-old mice and tail vein injection in 2-week-old mice. We have rewritten the sections on injection methods in the revised manuscript.

b) In line 145, the authors state that ABE1 only contains one copy of sgRNA, while ABE2 has two sgRNAs. However, in Figure 2D, ABE1 contains two copies of sgRNA in Cas9n-N, and ABE2 contains three copies of sgRNA. This inconsistency between the main text, figure legends, and the figure itself could cause confusion for readers and researchers trying to understand the methodology. It is crucial to correct this inconsistency and provide accurate information on copy numbers for sgRNAs used in each AAV construct.

Response: Thank you for raising this issue. We have revised the manuscript to clarify the fact that ABE1 contains two copies of sgRNA in Cas9n-N, whereas ABE2 contains three copies of sgRNA.

c) To support the results, the manuscript should include a clear method for quantifying Western blot bands. For example, in Figure 3H, the dystrophin expression of the ABE2 group appears higher than 50% of the wild type (WT), even after normalization to Vinculin expression. However, Figure 3I shows that ABE2 only has approximately 40% dystrophin expression compared to WT. Including a description of the Western blot band quantification method would enhance the transparency and reliability of the results.

Response: We have included a description of the Western blot band quantification method in the revised manuscript as follow.

“The statistical analysis of Western blotting band quantification was performed as follow. In brief, the values of dystrophin band intensity for the 50%, 25%, 10% of WT muscle protein lysate, Control-, ABE1- and ABE2-treated muscle lysates were measured using ImageJ software. Dystrophin band intensity was normalized to that of Vinculin band as internal loading control. Then, dystrophin level of control or treated DMD mice was further normalized by that of the WT mice. The results were presented as the percentage of the wildtype dystrophin level for the individual lane.”

Minor comments:

In Figure 1F, the muscle tissue type should be labeled to provide clarity.

Response: We have labeled the gel with muscle tissue type as suggested to provide clarity (Fig. R13).

Fig. R13: Western blot showed the absence of dystrophin in the indicated muscle tissues.

All Western gels and PCR gels should include molecular weight markers for accurate size estimation. Full blot image of Western blot should be presented in Suppl.

Response: We have revised figures with the western blot and PCR gels results as suggested to include molecular weight markers or labels for accurate size estimation. Full blot images were also provided as commented in the supplementary files.

Figure 4F is missing the mice type label, which should be added for clarity.

Response: We have added the missing label of mouse type in the revised Fig. 4f as follow (Fig. R12).

Fig. R12: Immunohistochemistry for dystrophin in TA, DI, and heart of DMD mice was performed 6 weeks after IP injection. Dystrophin is shown in green. Scale bar, 200 μ m.

In Figure 5C, "teatement" is labeled on molecular markers. Please fix it.

Response: Thank you for the careful review. We have fixed the inaccurate label as follow (Fig. R14).

Fig. R14: RT-PCR products from muscle of DMD mice with or without ABE2 treatment were analyzed by gel electrophoresis.

The figure legends in supplemental Figure S6 and S9 do not match the figure panels. Please ensure that the figure legends accurately describe the content of the figures.

Response: We felt sorry for the mistake. The manuscript was revised as commented.

Other suggestion (optional)

CK levels should be measured in the systemic delivery mice, but this reviewer understand that the blood samples might not be available anymore since the experiments already finished.

Response: We have provided the CK measurement results for DMD mice with 6-week systemic treatment in the Fig. S11 of our previous manuscript. In the revised study, we also included the CK measurements of DMD mice treated for both 6 weeks and 10 months presented as follow (**Fig. R15**).

Fig. R15: CK activity was measured 6-week and 10-month after intravenous injection of ABE2 viral particles.

Reviewer #4 (Remarks to the Author):

This manuscript reports a novel humanized mouse model of DMD and an adenine base editor (ABE) strategy to induce exon skipping. They show a remarkable reconstitution of dystrophin expression depending on the construct design.

Some limitations should be addressed:

1. Overall, there are many gene editing approaches in DMD models. What is the particular advantage and novelty compared to e.g. PMID: 33931459?

Response: Thanks for raising this issue. Overall, our study introduces several innovative aspects in the field of Duchenne Muscular Dystrophy (DMD) research. Firstly, it recognizes the significant differences between human and mouse DMD gene sequences. The generation of a humanized DMD mouse model is critical for evaluating the preclinical efficacy of drugs within mice. We successfully created a humanized DMD model by one-step knockout of mouse DMD exons 50 and 51, and insertion of human exon 50. This model exhibited pathological characteristics consistent with DMD patients.

Secondly, we efficiently achieved DMD exon 50 skipping through the optimization of the AAV-ABE vector and inteins split strategy. This was effective regardless of whether it was delivered via intramuscular injection or systemic delivery.

Thirdly, while previous studies have reported the application of ABE/CBE to modify DMD splice sites in mice (PMCID: PMC8087404/PMID: 33931459) and human cardiomyocytes (PMCID: PMC9792405), they only implemented it through intramuscular injection of TA muscle or in vitro assay, but did not evaluate the recovery of motor function and serum biomarkers via systemic delivery. In contrast, our manuscript systematically evaluates the therapeutic effects of the gene editing tool through both intraperitoneal and tail vein injections in the DMD model, and tracks the exon 50 skipping up to 10 months, demonstrating the long-term effectiveness of strategy via base-editing mediated exon-skipping.

These innovative aspects make this manuscript highly significant in the field of DMD research. It not only increases our understanding of the differences between human and mouse DMD gene sequences but also provides a reliable humanized DMD mouse model for more accurate evaluation of drug development. Additionally, the efficient exon 50 skipping achieved through AAV-ABE optimization offers a new approach for gene editing therapy in DMD. Most importantly, the manuscript extensively investigates the long-term effectiveness of base-editing, providing strong support for future clinical applications.

2. The methods section states in line 592 that "All animal experiments were performed and approved by the Institutional Animal Care and Use Committee (IACUC) of HuidaGene Therapeutics Inc., Shanghai, China." I assume that this statement is not correct as the IACUC probably did not perform the animal experiments. Furthermore, it is unusual that a company board approves animal experiments and not governmental or university bodies. In addition, no reference to international guidelines of animal experiments are made.

Response: Thank you for commenting on the issue. We have investigated the national

experimental and ethical regulations on the animal care and use for the company. The company's "Experimental Animal Use License" is regulated and issued by the Shanghai Science and Technology Commission (City Science and Technology Commission), and only is granted the "Experimental Animal Use License" after being reviewed and approved by the Science and Technology Commission of Shanghai. The City Science and Technology Commission conducts annual review of the license management, and the annual review includes facility operation, animal feeding, ethical review and other work related to animal experiments. After the annual inspection, the seal is stamped and the validity of the license is extended. Animal experiments comply with the following regulations and guide principles regarding animal management and welfare according to the AAALAC guidelines listed in the book "Guidelines for the Management and Use of Experimental Animals", compiled by the National Research Council (United States) as well as the "Regulations on the Management of Experimental Animals" of the National Science and Technology Commission of the People's Republic of China. Therefore, we have revised the section on study approval in the manuscript as follow to clarify the issue for ethical regulations.

"...All animal experiments were performed in HuidaGene Therapeutics Inc., Shanghai, China and approved by the Institutional Animal Care and Use Committee (IACUC) of the Shanghai Science and Technology Commission on a regular basis...."

3. Animal numbers are very small (n=3) raising statistical concerns. In addition, display of SEM instead of standard deviations is not correct.

Response: Thanks for the suggestion. The graph actually present data as mean \pm SD, and the manuscript has been revised.

4. Information regarding the cardiac phenotype is lacking, e.g. left ventricular function. Have authors characterized the heart in their model at baseline and in the gene therapy studies?

Response: DMD is a genetic disease caused by a lack or defect of the muscle dystrophin protein, leading to gradual muscle degeneration and weakening of muscle strength. The muscle degeneration also includes that of heart muscles, causing the heart function to gradually deteriorate, eventually leading to heart failure (PMID: 31147635). However, DMD mice demonstrate a certain compensatory ability in terms of heart function. Research has found that a protein called Utrophin exists in DMD mice, which can partially replace the missing dystrophin, thereby alleviating damage to heart function. The presence of Utrophin makes the heart muscles of DMD mice relatively healthy and the heart function relatively normal (PMID: 9288752). Cardiac ultrasound results show that in our treated and untreated humanized DMD mice versus wild-type mice, there were no significant changes in various cardiac indicators, except for a certain degree of increase in EF and FS in 10-month-old mice shown as follow (**Fig. S16**).

Fig. R16: Echocardiography was used to assess the cardiac function of mice after systemic delivery of ABE2. a-b, Representative echocardiographic images DMD mice with or without ABE2 administration were monitored for 6 weeks (a) and 10 months (b). Age-matched wild-type and DMD mice were included as controls. c, Echocardiographic analysis was performed in WT, DMD-mock, and DMD mice treated with ABE2 after 6 weeks and 10 months injection. LVID;d or LVID;s: Left Ventricular Internal Diameter during diastole or systole; LVPW;d or LVPW;s: Left Ventricular Posterior Wall Thickness during diastole or systole; LVPW;d or LVPW;s: Left Ventricular Posterior Wall Thickness during diastole or systole; LVAW;d or LVAW;s: Left Ventricular Anterior Wall Thickness during diastole or systole; LV Vol;d or LV Vol;s: Left Ventricular Volume during diastole or systole; EF: Ejection Fraction; FS: Fractional Shortening; CO: Cardiac Output; LV Mass (corrected): Left Ventricular Mass corrected for body surface area.

5. How do you explain the striking difference in efficiency between ABE1 and ABE2?

Response: Previous studies have shown that increasing the copy number of gRNA can enhance gene editing efficiency. A higher number of gRNA copies could generate more gRNA molecules that bind to the Cas9 enzyme, increasing the activity of Cas9 on the target

gene and thereby improving gene editing efficiency (PMID: 32128412, PMID: 26987018 and PMID: 28931002).

Reviewers' Comments:

Reviewer #1:

Remarks to the Author:

REVIEWER COMMENTS

Reviewer #1 (Remarks to the Author):

Has the new mouse model been characterized before or after the age of 8 weeks (the age where analyses were performed, legend Fig. 1). Please comment on the reasons why this time point was chosen. The presence and abundance of revertant fibers should be commented on.

Response: Thank you for the good comment. We have performed phenotypic characterization of the humanized mouse model at 2-week, 8-week and 24-week. We only showed the results of 8-week DMD mice in Fig.1. In the revised manuscript, we have provided all results shown as follow (Fig. R1) for comprehensive phenotypic characterization of DMD mice at different time points. Briefly, the muscular histology of the DMD mice showed the inward migration of cell nuclei in all the tested muscle tissues occurred as early as 2 weeks old (Fig. R1a). As the mice grew older (8 weeks and 24 weeks old), the populations of centrally nucleated fibers (CNFs) significantly increased and the inflammatory cell infiltration also appeared at 2 weeks of age, with the increasing severity over time (Fig. R1b). The appearance of creatine kinase (CK) in blood has been considered as a biochemical marker of muscle necrosis. Consistent with dystrophin deficient DMD patients, serum CK activity of this humanized DMD mice had a dramatic elevation at 2 weeks of age when compared to the wildtype (WT) controls, indicating the severity of muscle damage (Fig. R1c). Moreover, the DMD mice exhibited significant reduction in muscle strength as early as 6 weeks old when compared to the age-matched WT mice (Fig. R1d), recapitulating progressive muscle weakness of DMD patients. Therefore, 8-week-old DMD mice were chosen for evaluation. In the humanized DMD mice, we detect very few revertant fibers that are difficult to be distinguished from background antibody staining signal (Fig. R2).

Reviewer comment: The figure is labeled S1 in the manuscript, not R1. The additional results are important. The alignment of mouse and human sequence is helpful. The yellow arrowheads in S1b occlude parts of the sections and should be removed. The histology does not match the bar graphs (S1c). If 80% central nuclei are indicated in the bar graph then they should be present in histology as well. Please select representative images and indicate how many fibers were counted for analyzing the centrally nucleated fibers. Heart muscle fibers, physiologically, have a single (rarely two) centrally located nucleus. If nothing else is pointed out in the histologically features, the histology of heart sections can be removed. Fig S1d: CK levels: There is quite a literature on creatine kinase levels (U/l) in mice \pm muscular dystrophy. One classical citation is: Morgan et al. Res Commun Chem Pathol Pharmacol PMID: 7221187. The level in wildtype mice do not exceed 150-200 U/l. Please comment on values of 1000 and above. The authors state in the figure below that the data are presented as mean \pm SEM. The now submitted manuscript states it is the mean \pm SD. Which one is true?
Please quantify revertant fibers.

Histology Fig. 1: Severe fibrosis is not shown in the TA.

Response: Thanks for your comment. Through Sirius red and HE staining, we observed severe fibrosis in the diaphragm and anterior tibialis muscles. The H&E and Sirius red staining results of 8 DMD mice are presented as follow (Fig. R3), and representative results have been selected to replace Fig.1e.

Reviewer: The now selected image seems more fitting. Fine.

Fig. 1H: The mean and SEM of CK levels do not reflect the data points shown.

Response: Thanks for raising this issue. We have replotted the results (Fig. R4) as suggested to revise the mistake.

Response Reviewer#1: The mean appears correct now. The figure legend of the figure in the second submission describes, contrary to the original submission file, that the SD is plotted. However, we would expect the SD error bars to be much larger in this case because two values are higher than the other four. If the authors intend to show the SD, please replot the figure.

Discuss differences between human and murine exon 50. In which aspect is the model humanized?

Response: Although the homology between mouse and human DMD genes is very high, there are still significant sequence differences near the splicing site (Fig. R5). To generate a genetically humanized DMD mouse model carrying specific human exon deletion mutations, we knocked in human exon 50 with flanking 200 bp sequences to replace mouse exons 50 and 51 in a single step. This will facilitate the study of gene-editing-mediated exon skipping therapy.

Fig. R5: Alignment of human and mouse exon 50 sequence. Human exon 50 (Yellow labeling), mouse exon 50 (Blue labeling), sgRNA (magenta line), Protospacer adjacent motif (Red box).

Response Reviewer#1: Is this Fig. S1a? We appreciate that the comparison of the mouse and human sequences is now included.

Fig. 3C: Please comment on the additional band in the control-panel. The band demonstrates that you see exon 51 KO and in addition exons 50 plus 51 KO.

Response: Thanks for raising this issue. We sequenced the unexpectedly appearing band and found that this sequence was generated by exon skipping. We speculate that it is generated by spontaneous exon skipping in mice.

Response Reviewer#1: OK.

Fig. 3B shows an A>G conversion of 20% for ABE1 and 25% for ABE2. How do the authors then explain 80 – 100% RNA skipping efficiency? Are there bystander edits? Please provide a chromatogram of the edited and unedited splice donors sites.

Response: Thanks for your good comment. We also find the discrepancy between genome editing and RNA skipping efficiency intriguing. Previous study (PMID: 34698513 , PMID: 30854433) reported similar results with us, which might be due to different stability of edited and unedited transcripts or methodologic difference between DNA and RNA editing analysis. For bystander editing analysis, we provided the reads analysis presented as follow (Fig. R6) to show alleles with or without bystander editing events.

Fig. R6 : Deep-seq reads analysis of ABE1- and ABE2-edited DMD gene in DMD mice.

Response Reviewer#1: Bystander editing rates are now shown. The discrepancy between the editing rates and RNA skipping efficiency should be discussed in the manuscript?

Fig. 4B and D: Please show bars on the same scale from 1 to 100% (same for 5B and D)

Response: Thanks for the suggestion. The Fig. 4B and D, Fig. 5B and D have been modified as suggested (Fig. R7).

Response Reviewer#1: The adjustments of the scales of 4D and the figure now called 5C is good. Here, the scale is now 0-100% as suggested. For 4B and 5B, however, the visualisation is less clear than before and harder to compare with the color-coded heat-map. Please find a way to show bar graphs side by side.

Also the following improvements should be made:

A) Addition of a label of the Y-Axis in the graphs of 4B/5B (0-20 ---- unit missing).

B) How come the editing efficiency now seems to have dropped to 20% as compared to the original submission.

Now in the second submission:

Fig. R7: Genome editing and RNA skipping efficiency after intraperitoneal administration of AAV-ABE2. a, Genome editing efficiency heatmap in heart, DI and TA for control and treated mice. b, Percentage of RNA skipping in heart, DI and TA for control and treated mice. n = 3. Data are represented as mean \pm SD. Each dot represents an individual mouse.

Fig. 4C and 5C: The sizes of the PCR products are not comprehensible. Please list all primers in the Supplement or name, if a different marker was used. A scheme of the size of the expected PCR products would ease the reading of the manuscript.

Response: Thank you for your careful review and helpful suggestions. The sizes of the PCR products were added in Fig. 4C and 5C shown as follow (Fig. R8), and all primers were also provided in Supplementary Table S1 of the revised manuscript.

Fig. R8: RT-PCR products from muscle of DMD mice were analyzed by gel electrophoresis. 449 bp and 340 bp bands are from transcripts without and with exon 50 skipping respectively.

Response Reviewer#1: It is now easier to follow the sizes of the PCR products with labelled PCR product sizes and marker lanes.

Fig. S3: Please describe how the off-target analysis was performed. How many mismatches were allowed? How were the eight sites selected? Which PAM-specification was used for the prediction. The (Cas)-OFFinder is not cited (Bae et al, 2014).

Response: Thank you for your comment. We have revised the method description for off-target analysis. Using the PAM sequence (5'-NG-3') of XCas9 3.7 (TLIKDIV SpCas9) from *Streptococcus pyogenes* for prediction on the Cas-offinder website, all 14 potential off-target sites with 3 mismatched positions are selected for evaluation. In the revised manuscript, we have included all 14 predicted off-target sites with 3 mismatches and analyzed them using deep sequencing, showing high on-target editing with sgRNA6 but undetectable off-target editing events as follow (Fig. R9). In addition, we have cited the relevant paper as suggested.

Fig. R9: Off-target analysis of ABE2-mediated base editing for DMD gene. a, Alignment of the top 14 off-target sites in human genome. The target adenine (A7) is colored red. b, Percentages of adenine editing in the all 14 off-target sites. Dots and bars represent biological replicates and data are presented as means \pm SD (n = 3).

Response Reviewer#1: It is now included how off-target sites were selected with transparent inclusion criteria. Please adjust the figure legend describing in line 959 the alignment of the top TEN off-targets. It is 14.

There is no mentioning of skipping in the muscle stem cells, satellite cells.

Response: Thanks for raising this issue. Dystrophin is expressed in differentiated myofibers and activated muscle stem cells (PMCID: PMC4839960). After intravenous injection of DMD mice with AAV-ABE2 for 6 weeks, we dissected the tibialis anterior muscle and isolated satellite cells for deep sequencing and RT-PCR analysis. The results showed that satellite cells have comparable A-to-G editing efficiency of up to 30% as the tibialis anterior muscle (Fig. R10). However, there is a low

amount of mouse satellite cells, making it difficult to extract RNA and perform effective gel electrophoresis analysis of RNA skipping rate. Previous studies have reported that by using adeno-associated virus (AAV) to deliver gene editing tools targeting the DMD mutation region, it can be effectively delivered to muscle stem cells, allowing for gene editing and restoring their normal differentiation and renewal functions (PMID: 36995603, PMC4924477). Indeed, we think that your suggestion is very important and have taken it seriously in our lab to optimize satellite cells protocol for RNA skipping evaluation and hopefully get it done in another study.

Fig. R10: Gene editing analysis in satellite cells of DMD mice. a, Satellite cells of DMD mice treated with AAV-ABE2 showed PAX7 expression. b, Gene editing efficiency measured with deep-seq for satellite cells of DMD mice treated with AAV-ABE2. c, Deep-seq reads analysis for ABE2-edited DMD gene in satellite cells.

Response Reviewer#1: How were satellite cells isolated? Please provide the protocol in the Methods section. Have other myogenic markers been stained? The immunofluorescent picture that is provided does not clearly show nuclei positive for Pax7. The magnification is insufficient and the morphology of the cells do not fit.

How long does the effect of skipping last? Was six weeks after editing the only timepoint that was checked? Please clarify how i.v. delivery was performed: retro-orbital or via tail vein?

Response: To examine the long-term therapeutic effect of ABE2 treatment, DMD mice with or without ABE2 administration were monitored for 10 months and then euthanized to analyze muscle tissues. Our results revealed durable therapeutic efficacy of ABE2 administration for 10 months, which are added in the revised manuscript presented as follow (Fig. R11).

Fig. R11: Intravenous delivery of ABE system efficiency rescues dystrophin expression and muscle function in humanized DMD mice. a, Schematic of intravenous administration of ABE2 particles. Tissues were collected for genomic DNA, RNA, immunoblotting and immunofluorescence experiments at 6 weeks (n=3) and 10 months (n=6) after treatment. Black arrows indicate time points for tissue collection after IV injection. b, Measurement by deep sequencing of splicing site editing efficiency in TA, DI, and heart after systemic delivery of ABE2. RT-PCR products from muscle of DMD Δ mE5051,KIhE50/Y mice were analyzed by deep sequencing. c, RNA exon-skipping efficiency. d, Immunohistochemistry for dystrophin in TA, DI, and heart of DMD Δ mE5051,KIhE50/Y mice was performed 6 weeks or 10 months after intravenous injection. Dystrophin is shown in green. Scale bar, 200 μ m. e-f, Western blot analysis shows restoration of dystrophin expression in the TA, DI, and heart of DMD Δ mE5051,KIhE50/Y mice 6 weeks or 10 months after injection. Dilutions of protein extract from WT mice were used to standardize dystrophin expression (10%, 25%, and 50%). Vinculin was used as the loading control. Forelimb grip strength (g) and rotarod rod performance (h) were measured two days in WT, and DMD Δ mE5051,KIhE50/Y mice treated without or with ABE2 particles. i, The remaining strength was measured during 10 repetitions at 10-second intervals. Dots and bars represent biological replicates and data are presented mean \pm SD. Significance is indicated by asterisks and was determined in Fig. 5g, h using unpaired two-tailed Student's t test or in Fig. 5i using ANOVA multiple comparison test.

Response Reviewer#1: The 10 month timepoint is a valuable addition to the manuscript.

The claims in the discussion should be down-sized. It has not been examined whether gene editing byproducts were produced. Also, whether this is a safe alternative to existing strategies is also questionable. The topic of off-target analysis is discussed only very briefly. The specific ABE that was used is not discussed. The very similar paper of Chemello et al. 2021; DOI: 10.1126/sciadv.abg4910 has not been cited. The major difference to the paper by Chemello et al. is the fact that there was no humanized exon 50 in the Chemello-paper.

Response: Thanks for helping improve our study. We have revised our manuscript as suggested

and cited relevant papers.

Response Reviewer#1: The Chemello paper is now mentioned including an explanation of what is unique to this manuscript. The sentence that the ABE strategy is safe was downsized.

Minor:

Remove Company names from the legend

Response: Thank you for your careful review and helpful suggestions. Company names in the legend were removed as suggested.

Fig. 4F: Please add a legend to the columns either in the figure itself or in the legend text.

Response: The legend text has been added to the figures of our revised manuscript.

Fig. R12: Immunohistochemistry for dystrophin in TA, DI, and heart of DMD mice was performed 6 weeks after IP injection. Dystrophin is shown in green. Scale bar, 200 μm .

Fig. S6: The legend does not correspond to the figure.

Response: We felt sorry for the mistake. The manuscript was revised as commented.

Response Reviewer#1: Ok. The legend of the now called Figure S7B still does not seem to reflect what is shown in the figure.

Please explain why editing efficiency can be higher than 100%.

also in S3:

Reviewer #2:

Remarks to the Author:

Reviewer #3:

Remarks to the Author:

Majority of the concerns raised by this reviewer and co-reviewer have been addressed by the authors.

Minor comments on the raw Western blot images.

1. The raw blot image for Fig.1e-TA-Dystrophin did not match the one in the main text. Please attach the correct one.

2. Figure panels for the raw images are mislabelled. Western blot gel images in the main text are shown in Fig 1e, 3h, 4e, 5d, and 5e. Please fix it.

Reviewer #4:

Remarks to the Author:

Thank you for your detailed answers.

Regarding the cardiac phenotype, the authors provided echocardiography data showing a trend towards an increased EF in DMD mice with a significant reduction in the gene therapy group (S16). Could this be attributed to the statistical variation considering very low animal numbers (n=3 in WT and control mice, n=5 in the treatment group)?

Likewise, animal numbers are very low (n=3) in many other analyses of the gene therapy study. The authors have not addressed this issue.

Minor issues:

-Please revise grammar. There are still some flaws.

-Correct Fig 4a. "Intraperitoneal"

Reviewers' comments:

We greatly appreciate all insightful comments of reviewers in helping improve our study. We believe that these revisions address the concerns raised by the reviewers and strengthen the overall quality of the study. We are confident that the revised manuscript makes a significant contribution to the field and hope that the reviewers will find the changes satisfactory. Our response were highlighted in blue in response letter below and revisions in manuscript were marked in yellow.

Reviewer #1 (Remarks to the Author):

REVIEWER COMMENTS

Reviewer #1 (Remarks to the Author):

Has the new mouse model been characterized before or after the age of 8 weeks (the age where analyses were performed, legend Fig. 1). Please comment on the reasons why this time point was chosen. The presence and abundance of revertant fibers should be commented on.

Response: Thank you for the good comment. We have performed phenotypic characterization of the humanized mouse model at 2-week, 8-week and 24-week. We only showed the results of 8-week DMD mice in Fig.1. In the revised manuscript, we have provided all results shown as follow (Fig. R1) for comprehensive phenotypic characterization of DMD mice at different time points. Briefly, the muscular histology of the DMD mice showed the inward migration of cell nuclei in all the tested muscle tissues occurred as early as 2 weeks old (Fig. R1a). As the mice grew older (8 weeks and 24 weeks old), the populations of centrally nucleated fibers (CNFs) significantly increased and the inflammatory cell infiltration also appeared at 2 weeks of age, with the increasing severity over time (Fig. R1b). The appearance of creatine kinase (CK) in blood has been considered as a biochemical marker of muscle necrosis. Consistent with dystrophin deficient DMD patients, serum CK activity of this humanized DMD mice had a dramatic elevation at 2 weeks of age when compared to the wildtype (WT) controls, indicating the severity of muscle damage (Fig. R1c). Moreover, the DMD mice exhibited significant reduction in muscle strength as early as 6 weeks old when compared to the age-matched WT mice (Fig. R1d), recapitulating progressive muscle weakness of DMD patients. Therefore, 8-week-old DMD mice were chosen for evaluation. In the humanized DMD mice, we detect very few revertant fibers that are difficult to be distinguished from background antibody staining signal (Fig. R2).

Response Reviewer#1: The figure is labeled S1 in the manuscript, not R1. The additional results are important. The alignment of mouse and human sequence is helpful. The yellow arrowheads in S1b occlude parts of the sections and should be removed. The histology does not match the bar graphs (S1c). If 80% central nuclei are indicated in the bar graph

then they should be present in histology as well. Please select representative images and indicate how many fibers were counted for analyzing the centrally nucleated fibers. Heart muscle fibers, physiologically, have a single (rarely two) centrally located nucleus. If nothing else is pointed out in the histologically features, the histology of heart sections can be removed. Fig S1d: CK levels: There is quite a literature on creatine kinase levels (U/l) in mice \pm muscular dystrophy. One classical citation is: Morgan et al. Res Commun Chem Pathol Pharmacol PMID: 7221187. The level in wildtype mice do not exceed 150-200 U/l. Please comment on values of 1000 and above. The authors state in the figure below that the data are presented as mean \pm SEM. The now submitted manuscript states it is the mean \pm SD. Which one is true? Please quantify revertant fibers.

Response: We appreciate the valuable feedback provided by the reviewer. In response to the concerns raised, we have made the following revisions to the manuscript:

Figure Labeling: We acknowledge the oversight in labeling of the Fig. R1 instead of Fig. S1 in the manuscript. We recognize the importance of the additional results and have ensured that they are appropriately highlighted in the revised manuscript.

Fig. S1b: We have removed the yellow arrowheads in Fig. S1b to ensure that the sections are clearly visible in the revised figure.

Histology and Bar Graphs (**Fig. R1**): We have addressed the discrepancy between the histology and the bar graphs in Fig. S1c by selecting representative images and providing a detailed description of the fiber counting method to ensure accuracy in the analysis of centrally nucleated fibers as follow. Additionally, the histology of heart sections was removed as suggested.

“...The percentage of central nucleation fibers is obtained by counting the number of intra-nuclear fibers in whole-muscle scanning of GA, DI, and QA muscles...”

Fig. R1 (related to Fig. S1c, d): Histological analysis of DMD mice (a) and centrally nucleated fibers statistics (b)

CK Levels (**Fig. R2**): We have conducted a thorough literature review (**Table R1**) on creatine kinase levels in mice with muscular dystrophy, including the classic citation by Morgan et al. (PMID: 7221187). Previous studies also reported varied CK activity level (**Table R1**). The differences for CK values of \sim 1000 in our wildtype mice from that in Morgan et al. paper may be attributed to different experimental and normalization methods. In

addition, Morgan et al. study used Bar Harbor 129 ReJ strain that might be also different from C57/B6 mice used by us. In our study, we measured the CK activity with methods adapted from Sanchez-Castro et al. study (PMID: 33633730).

Fig. R2 (related to Fig. S1d): Time-course measurement of creatine kinase activity in DMD mice

Table R1: CK activity results reported in previous study

	Background	CK(U/liter)	References
	WT	327.00±118.90	
1	Dmd exon23 carried nonsense mutation (mdx)	7985.00±1825.96	PMID: 28439558
2	WT mdx	318 8834.60±3928.75	PMID: 25123483
3	WT mdx	~300 ~14000	PMID: 26721683
4	WT Mice with a 4-bp deletion at Dmd exon 4 (DmdE4*)	~250 ~2500	PMID: 34698513
5	WT mdx	223.71±85.69 4744.25±2430.92	PMID: 31591596
6	WT Dmd exon44 deletion (Δ Ex44)	~3000 ~70000	PMID: 30854433
7	WT	~1000	PMID: 33633730
8	Normal mice Dystrophic mic	24-102 79-157	PMID: 7221187

Data Presentation: We have clarified that data are presented as mean +/- s.d. as stated in the figure legend and methods section.

Revertant fibers (**Fig. R3**): The number of revertant fibers was quantified in the revised manuscript.

Fig. R3 (related to Fig. S1b): Revertant fibers analysis in DMD mice with different ages

Histology Fig. 1: Severe fibrosis is not shown in the TA.

Response: Thanks for your comment. Through Sirius red and HE staining, we observed severe fibrosis in the diaphragm and anterior tibialis muscles. The H&E and Sirius red staining results of 8 DMD mice are presented as follow (Fig. R3), and representative results have been selected to replace Fig.1e.

Reviewer: The now selected image seems more fitting. Fine.

Fig. 1H: The mean and SEM of CK levels do not reflect the data points shown.

Response: Thanks for raising this issue. We have replotted the results (Fig. R4) as suggested to revise the mistake.

Response Reviewer#1: The mean appears correct now. The figure legend of the figure in the second submission describes, contrary to the original submission file, that the SD is plotted. However, we would expect the SD error bars to be much larger in this case because two values are higher than the other four. If the authors intend to show the SD, please replot the figure.

Response: We appreciate reviewer's feedback regarding the figure legend and the representation of standard deviation (SD) error bars. We have ensured that this number is represented accurately with appropriate SD error bars and redraw the graph to reflect the expected representation of SD. It was fixed as commented.

Discuss differences between human and murine exon 50. In which aspect is the model humanized?

Response: Although the homology between mouse and human DMD genes is very high, there are still significant sequence differences near the splicing site (Fig. R5). To generate a genetically humanized DMD mouse model carrying specific human exon deletion mutations, we knocked in human exon 50 with flanking 200 bp sequences to replace mouse exons 50 and 51 in a single step. This will facilitate the study of gene-editing-mediated exon skipping therapy.

Fig. R5: Alignment of human and mouse exon 50 sequence. Human exon 50 (Yellow labeling), mouse exon 50 (Blue labeling), sgRNA (magenta line), Protospacer adjacent motif (Red box).

Response Reviewer#1: Is this Fig. S1a? We appreciate that the comparison of the mouse and human sequences is now included.

Response: We appreciate your attention to this detail and are pleased that this addition has been recognized.

Fig. 3C: Please comment on the additional band in the control-panel. The band demonstrates that you see exon 51 KO and in addition exons 50 plus 51 KO.

Response: Thanks for raising this issue. We sequenced the unexpectedly appearing band and found that this sequence was generated by exon skipping. We speculate that it is generated by spontaneous exon skipping in mice.

Response Reviewer#1: OK.

Response: We appreciate your attention to this detail and are pleased that this addition has been recognized.

Fig. 3B shows an A>G conversion of 20% for ABE1 and 25% for ABE2. How do the authors then explain 80 – 100% RNA skipping efficiency? Are there bystander edits? Please provide a chromatogram of the edited and unedited splice donors sites.

Response: Thanks for your good comment. We also find the discrepancy between genome editing and RNA skipping efficiency intriguing. Previous study (PMID: 34698513 , PMID: 30854433) reported similar results with us, which might be due to different stability of edited and unedited transcripts or methodologic difference between DNA and RNA editing analysis. For bystander editing analysis, we provided the reads analysis presented as follow (Fig. R6) to show alleles with or without bystander editing events.

Fig. R6: Deep-seq reads analysis of ABE1- and ABE2-edited DMD gene in DMD mice.

Response Reviewer#1: Bystander editing rates are now shown. The discrepancy between the editing rates and RNA skipping efficiency should be discussed in the manuscript?

Response: As suggested, we added discussion on the discrepancy between the editing rates and RNA skipping efficiency in the revised manuscript as follow.

“This discrepancy has been reported in previous studies (PMID: 26721684; PMID: 30166439; PMID: 29805845; PMID: 32892813; PMID: 34698513) as summarized below (Table R2). The discrepancy is likely due to the presence of multiple cell types within the muscle tissue, including endothelial cells, pericytes, macrophages, fibro-adipogenic progenitors, and potentially other cell types that are not yet well understood at this time (PMID: 29305000). Additionally, nonsense-mediated decay (NMD) may influence the

abundance of non-edited cDNA products, resulting in an apparent higher proportion of edited cDNA (PMID: 27259145). It is worth noting that, given the multinucleated nature of many murine muscles, even editing a single nucleus in a cardiomyocyte or skeletal muscles would result in the entire multinucleated cell regaining dystrophin protein, leading to a higher number of Dys+ cells compared to the relatively modest DNA base editing efficiency (PMID: 27259145).”

Table R2: Efficiency of DNA editing, RNA skipping and restored Dys+ fibers reported in previous study

	DNA	RNA	Dys+ fibers	WB	References
1	~2%	59%	67%	~8%	PMID: 26721684
2	2~4%	73.19%	-	60%	PMID: 30166439
3	5.4%	-	Estimated 60%	-	PMID: 29805845
4	Not provided	10.3%+4.3%	~72%	~50%	PMID: 32892813
5	3.9±1.3%	59.98±4.74%	86.9±10.1%	-	PMID: 34698513

Fig. 4B and D: Please show bars on the same scale from 1 to 100% (same for 5B and D)
 Response: Thanks for the suggestion. The Fig. 4B and D, Fig. 5B and D have been modified as suggested (Fig. R7).

Response Reviewer#1: The adjustments of the scales of 4D and the figure now called 5C is good. Here, the scale is now 0-100% as suggested. For 4B and 5B, however, the visualisation is less clear than before and harder to compare with the color-coded heatmap. Please find a way to show bar graphs side by side.

Response: We have used side-by-side display of bar graphs as suggested to improve clarity and facilitate comparison (**Fig. R4**). Your input is valuable to us, and we are committed to addressing this issue in the revised submission.

Fig. R4 (related to Fig. 4b): Heatmap and bar graph show base editing efficiency for DMD gene

Also the following improvements should be made:

- A) Addition of a label of the Y-Axis in the graphs of 4B/5B (0-20 ---- unit missing).
- B) How come the editing efficiency now seems to have dropped to 20% as compared to

the original submission.

Response: Thank you for the reviewer's feedback. We have replaced the heatmap graph with bar graph and added Y-axis label as commented. Editing efficiency results in the original submission was obtained by EditR analysis of sanger sequencing, which might be less precise than next-generation sequencing of amplicon that we used in the revised manuscript. The NGS analysis methods of editing efficiency have been thoroughly detailed in the methods section now.

Now in the second submission:

Fig. R7: Genome editing and RNA skipping efficiency after intraperitoneal administration of AAV-ABE2. a, Genome editing efficiency heatmap in heart, DI and TA for control and treated mice. b, Percentage of RNA skipping in heart, DI and TA for control and treated mice. n = 3. Data are represented as mean \pm SD. Each dot represents an individual mouse.

Fig. 4C and 5C: The sizes of the PCR products are not comprehensible. Please list all primers in the Supplement or name, if a different marker was used. A scheme of the size of the expected PCR products would ease the reading of the manuscript.

Response: Thank you for your careful review and helpful suggestions. The sizes of the PCR products were added in Fig. 4C and 5C shown as follow (Fig. R8), and all primers were also provided in Supplementary Table S1 of the revised manuscript.

Fig. R8: RT-PCR products from muscle of DMD mice were analyzed by gel electrophoresis. 449 bp and 340 bp bands are from transcripts without and with exon 50 skipping respectively.

Response Reviewer#1: It is now easier to follow the sizes of the PCR products with labelled PCR product sizes and marker lanes.

Response: Your acknowledgment of the enhanced visibility is greatly appreciated.

Fig. S3: Please describe how the off-target analysis was performed. How many mismatches were allowed? How were the eight sites selected? Which PAM-specification was used for the prediction. The (Cas)-OFFfinder is not cited (Bae et al, 2014).

Response: Thank you for your comment. We have revised the method description for off-target analysis. Using the PAM sequence (5'-NG-3') of XCas9 3.7 (TLIKDIV SpCas9) from *Streptococcus pyogenes* for prediction on the Cas-offfinder website, all 14 potential off-target sites with 3 mismatched positions are selected for evaluation. In the revised manuscript, we have included all 14 predicted off-target sites with 3 mismatches and analyzed them using deep sequencing, showing high on-target editing with sgRNA6 but undetectable off-target editing events as follow (Fig. R9). In addition, we have cited the relevant paper as suggested.

Fig. R9: Off-target analysis of ABE2-mediated base editing for DMD gene. a, Alignment of the top 14 off-target sites in human genome. The target adenine (A7) is colored red. b,

Percentages of adenine editing in the all 14 off-target sites. Dots and bars represent biological replicates and data are presented as means \pm SD (n = 3).

Response Reviewer#1: It is now included how off-target sites were selected with transparent inclusion criteria. Please adjust the figure legend describing in line 959 the alignment of the top TEN off-targets. It is 14.

Response: Thanks for the good comments. We have revised the typos in the manuscript.

There is no mentioning of skipping in the muscle stem cells, satellite cells.

Response: Thanks for raising this issue. Dystrophin is expressed in differentiated myofibers and activated muscle stem cells (PMCID: PMC4839960). After intravenous injection of DMD mice with AAV-ABE2 for 6 weeks, we dissected the tibialis anterior muscle and isolated satellite cells for deep sequencing and RT-PCR analysis. The results showed that satellite cells have comparable A-to-G editing efficiency of up to 30% as the tibialis anterior muscle (Fig. R10). However, there is a low amount of mouse satellite cells, making it difficult to extract RNA and perform effective gel electrophoresis analysis of RNA skipping rate. Previous studies have reported that by using adeno-associated virus (AAV) to deliver gene editing tools targeting the DMD mutation region, it can be effectively delivered to muscle stem cells, allowing for gene editing and restoring their normal differentiation and renewal functions (PMID: 36995603, PMC4924477). Indeed, we think that your suggestion is very important and have taken it seriously in our lab to optimize satellite cells protocol for RNA skipping evaluation and hopefully get it done in another study.

Fig. R10: Gene editing analysis in satellite cells of DMD mice. a, Satellite cells of DMD mice treated with AAV-ABE2 showed PAX7 expression. b, Gene editing efficiency measured with deep-seq for satellite cells of DMD mice treated with AAV-ABE2. c, Deep-seq reads analysis for ABE2-edited DMD gene in satellite cells.

Response Reviewer#1: How were satellite cells isolated? Please provide the protocol in the Methods section. Have other myogenic markers been stained? The immunofluorescent picture that is provided does not clearly show nuclei positive for Pax7. The magnification is insufficient and the morphology of the cells do not fit.

Response: The protocol of satellite cells isolation was provided in the methods section as follow. We have addressed the concern by incorporating the use of both Pax3 and Pax7 antibodies (**Fig. R5**). During the satellite cell isolation and culture process, the cells exhibit a predominantly round morphology, and gradually transition to a spindle-shaped morphology thereafter. The manuscript now includes the staining results of satellite cells after 4 days of isolation.

“The hindlimb and forelimb skeletal muscles were removed after the mice were euthanized. The muscles were washed twice with Dulbecco’s phosphate-buffered saline (DPBS) (Thermo Fisher Scientific) and finely chopped. Subsequently, they were digested using a 0.2% Collagenase type 2 (Gibco) solution for 60 minutes in a shaking water bath at 37°C. A second digestion was performed with a solution consisting of 0.2%

Collagenase type 2 and 0.4% Dispase (GIBCO) in Rinsing media for 30 minutes in a shaking water bath at 37°C. The digested tissue was then passed through a 40mm filter to collect the filtrate. In order to increase the probability of available satellite cells, a purification process was conducted by wall sticking screening twice, each time for 1 hour. The prepared cell suspension was then inoculated into ECM-coated (Sigma) coverslips. The satellite cells were cultured in growth medium (Ham's F10, 10% fetal bovine serum) supplemented with fibroblast growth factor (FGF) (Gibco) at 37°C in a humidified incubator with 5% CO₂."

Fig. R5 (related to Fig. S10a): Immunostaining results for isolated satellite cells

How long does the effect of skipping last? Was six weeks after editing the only timepoint that was checked? Please clarify how i.v. delivery was performed: retro-orbital or via tail vein?

Response: To examine the long-term therapeutic effect of ABE2 treatment, DMD mice with or without ABE2 administration were monitored for 10 months and then euthanized to analyze muscle tissues. Our results revealed durable therapeutic efficacy of ABE2 administration for 10 months, which are added in the revised manuscript presented as follow (Fig. R11).

Fig. R11: Intravenous delivery of ABE system efficiency rescues dystrophin expression and muscle function in humanized DMD mice. a, Schematic of intravenous administration of ABE2 particles. Tissues were collected for genomic DNA, RNA, immunoblotting and immunofluorescence experiments at 6 weeks (n=3) and 10 months (n=6) after treatment. Black arrows indicate time points for tissue collection after IV injection. b, Measurement by deep sequencing of splicing site editing efficiency in TA, DI, and heart after systemic delivery of ABE2. RT-PCR products from muscle of DMDΔmE5051,KIhE50/Y mice were analyzed by deep sequencing. c, RNA exon-skipping efficiency. d, Immunohistochemistry for dystrophin in TA, DI, and heart of DMDΔmE5051,KIhE50/Y mice was performed 6 weeks or 10 months after intravenous injection. Dystrophin is shown in green. Scale bar,

200 μm . e-f, Western blot analysis shows restoration of dystrophin expression in the TA, DI, and heart of DMD ΔmE5051 ,KlhE50/Y mice 6 weeks or 10 months after injection. Dilutions of protein extract from WT mice were used to standardize dystrophin expression (10%, 25%, and 50%). Vinculin was used as the loading control. Forelimb grip strength (g) and rotarod rod performance (h) were measured two days in WT, and DMD ΔmE5051 ,KlhE50/Y mice treated without or with ABE2 particles. i, The remaining strength was measured during 10 repetitions at 10-second intervals. Dots and bars represent biological replicates and data are presented mean \pm SD. Significance is indicated by asterisks and was determined in Fig. 5g, h using unpaired two-tailed Student's t test or in Fig. 5i using ANOVA multiple comparison test.

Response Reviewer#1: The 10 month timepoint is a valuable addition to the manuscript.

Response: We appreciate the reviewer's comments on the significance of our findings.

The claims in the discussion should be down-sized. It has not been examined whether gene editing byproducts were produced. Also, whether this is a safe alternative to existing strategies is also questionable. The topic of off-target analysis is discussed only very briefly. The specific ABE that was used is not discussed. The very similar paper of Chemello et al. 2021; DOI: 10.1126/sciadv.abg4910 has not been cited. The major difference to the paper by Chemello et al. is the fact that there was no humanized exon 50 in the Chemello-paper.

Response Reviewer#1: The Chemello paper is now mentioned including an explanation of what is unique to this manuscript. The sentence that the ABE strategy is safe was downsized.

Response: Thanks for helping improve our study. We have revised our manuscript as suggested and cited relevant papers.

Minor:

Remove Company names from the legend

Response: Thank you for your careful review and helpful suggestions. Company names in the legend were removed as suggested.

Fig. 4F: Please add a legend to the columns either in the figure itself or in the legend text.

Response: The legend text has been added to the figures of our revised manuscript.

Fig. R12: Immunohistochemistry for dystrophin in TA, DI, and heart of DMD mice was performed 6 weeks after IP injection. Dystrophin is shown in green. Scale bar, 200 μm .

Fig. S6: The legend does not correspond to the figure.

Response: We felt sorry for the mistake. The manuscript was revised as commented.

Response Reviewer#1: Ok. The legend of the now called Figure S7B still does not seem to reflect what is shown in the figure. Please explain why editing efficiency can be higher than 100%. also in S3:

Response: Thank you for the keen observation by reviewer#1. The legend of Fig. S7b was revised to reflect what is shown in the figure. For the editing efficiency presented in Fig. S3, we would like to clarify that the analysis was performed using relative editing efficiency and we normalized the data of intein-split ABE using the full-length ABE as reference and assigning mean value of full-length ABE to 100%. We have revised the sentence in the manuscript.

Reviewer #2 (Remarks to the Author):

Reviewer #3 (Remarks to the Author):

Majority of the concerns raised by this reviewer and co-reviewer have been addressed by the authors.

Response: We appreciate the reviewer's comments on the significance of our findings.

Minor comments on the raw Western blot images.

1. The raw blot image for Fig.1e-TA-Dystrophin did not match the one in the main text. Please attach the correct one.

2. Figure panels for the raw images are mislabelled. Western blot gel images in the main text are shown in Fig 1e, 3h, 4e, 5d, and 5e. Please fix it.

Response: Thanks for carefully reviewing our results. We have revised images in the manuscript.

Reviewer #4 (Remarks to the Author):

Thank you for your detailed answers.

Regarding the cardiac phenotype, the authors provided echocardiography data showing a trend towards an increased EF in DMD mice with a significant reduction in the gene therapy group (S16). Could this attributed to the statistical variation considering very low animal numbers (n=3 in WT and control mice, n=5 in the treatment group)? Likewise, animal numbers are very low (n=3) in many other analyses of the gene therapy study. The authors have not addressed this issue.

Response: We have now provided additional echocardiography data results for a larger number of mice with increased sample size of up to 6-8 animals per group than that in our

last submission. (Fig. R6). Furthermore, the manuscript has been updated to include the corresponding results of the echocardiography in the original submission. We believe that these revisions would address the concerns raised by the reviewer and strengthen the overall quality of the study.

Fig R6 (related to Fig. S16): Echocardiography was used to assess the cardiac function of mice after systemic delivery of ABE2.

a-b, Representative echocardiographic images for DMD Δ mE5051,KihE50/Y mice with or without ABE2 administration were monitored for 6 weeks (**a**) and 10 months (**b**). Age-matched wild-type and DMD mice were included as controls. **c**, Echocardiographic analysis was performed in WT, DMD-mock, and DMD mice treated with ABE2 after 6 weeks and 10 months injection. LVID;d or LVID;s: Left Ventricular Internal Diameter during diastole or systole; LVPW;d or LVPW;s: Left Ventricular Posterior Wall Thickness during diastole or systole; LVPW;d or LVPW;s: Left Ventricular Posterior Wall Thickness during diastole or systole; LVAW;d or LVAW;s: Left Ventricular Anterior Wall Thickness during diastole or systole; LV Vol;d or LV Vol;s: Left Ventricular Volume during diastole or systole; EF: Ejection Fraction; FS: Fractional Shortening; CO: Cardiac Output; LV Mass (corrected):

Left Ventricular Mass corrected for body surface area. Values are shown as mean \pm s.d (n=5 or 8). Significance is indicated by asterisk and determined using unpaired two-tailed Student's t test. NS represents not statistically significant.

Minor issues:

-Please revise grammar. There are still some flaws.

-Correct Fig 4a. "Intraperitoneal"

Response: Thanks for carefully reviewing our results. We have revised the typo and grammar in the manuscript.

Reviewers' Comments:

Reviewer #1:

Remarks to the Author:

Reviewers' comments:

We greatly appreciate all insightful comments of reviewers in helping improve our study. We believe that these revisions address the concerns raised by the reviewers and strengthen the overall quality of the study. We are confident that the revised manuscript makes a significant contribution to the field and hope that the reviewers will find the changes satisfactory. Our response were highlighted in blue in response letter below and revisions in manuscript were marked in yellow.

Reviewer #1 (Remarks to the Author):

The manuscript was co-reviewed by an Early Career Researcher together with a senior researcher. This is part of the Nature Communications initiative to facilitate training in peer review and to provide appropriate recognition for Early Career Researchers who co-review manuscripts.

REVIEWER COMMENTS

Reviewer #1 (Remarks to the Author):

Has the new mouse model been characterized before or after the age of 8 weeks (the age where analyses were performed, legend Fig. 1). Please comment on the reasons why this time point was chosen. The presence and abundance of revertant fibers should be commented on.

Response: Thank you for the good comment. We have performed phenotypic characterization of the humanized mouse model at 2-week, 8-week and 24-week. We only showed the results of 8-week DMD mice in Fig.1. In the revised manuscript, we have provided all results shown as follow (Fig. R1) for comprehensive phenotypic characterization of DMD mice at different time points. Briefly, the muscular histology of the DMD mice showed the inward migration of cell nuclei in all the tested muscle tissues occurred as early as 2 weeks old (Fig. R1a). As the mice grew older (8 weeks and 24 weeks old), the populations of centrally nucleated fibers (CNFs) significantly increased and the inflammatory cell infiltration also appeared at 2 weeks of age, with the increasing severity over time (Fig. R1b). The appearance of creatine kinase (CK) in blood has been considered as a biochemical marker of muscle necrosis. Consistent with dystrophin deficient DMD patients, serum CK activity of this humanized DMD mice had a dramatic elevation at 2 weeks of age when compared to the wildtype (WT) controls, indicating the severity of muscle damage (Fig. R1c). Moreover, the DMD mice exhibited significant reduction in muscle strength as early as 6 weeks old when compared to the age-matched WT mice (Fig. R1d), recapitulating progressive muscle weakness of DMD patients. Therefore, 8-week-old DMD mice were chosen for evaluation. In the humanized DMD mice, we detect very few revertant fibers that are difficult to be distinguished from background antibody staining signal (Fig. R2).

Response Reviewer#1: The figure is labeled S1 in the manuscript, not R1. The additional results are important. The alignment of mouse and human sequence is helpful. The yellow arrowheads in S1b occlude parts of the sections and should be removed. The histology does not match the bar graphs (S1c). If 80% central nuclei are indicated in the bar graph then they should be present in histology as well. Please select representative images and indicate how many fibers were counted for analyzing the centrally nucleated fibers. Heart muscle fibers, physiologically, have a single (rarely two) centrally located nucleus. If nothing else is pointed out in the histologically features, the histology of heart sections can be removed. Fig S1d: CK levels: There is quite a literature on creatine kinase levels (U/l) in mice \pm muscular dystrophy. One classical citation is: Morgan et al. Res Commun Chem Pathol Pharmacol PMID: 7221187. The level in wildtype mice do not exceed 150-200 U/l. Please comment on values of 1000 and above. The authors state in the figure below

that the data are presented as mean +/- SEM. The now submitted manuscript states it is the mean +/- SD. Which one is true? Please quantify revertant fibers.

Response: We appreciate the valuable feedback provided by the reviewer. In response to the concerns raised, we have made the following revisions to the manuscript:

Figure Labeling: We acknowledge the oversight in labeling of the Fig. R1 instead of Fig. S1 in the manuscript. We recognize the importance of the additional results and have ensured that they are appropriately highlighted in the revised manuscript.

Fig. S1b: We have removed the yellow arrowheads in Fig. S1b to ensure that the sections are clearly visible in the revised figure.

Histology and Bar Graphs (Fig. R1): We have addressed the discrepancy between the histology and the bar graphs in Fig. S1c by selecting representative images and providing a detailed description of the fiber counting method to ensure accuracy in the analysis of centrally nucleated fibers as follow. Additionally, the histology of heart sections was removed as suggested.

"...The percentage of central nucleation fibers is obtained by counting the number of intra-nuclear fibers in whole-muscle scanning of GA, DI, and QA muscles..."

Fig. R1 (related to Fig. S1c, d): Histological analysis of DMD mice (a) and centrally nucleated fibers statistics (b)

CK Levels (Fig. R2): We have conducted a thorough literature review (Table R1) on creatine kinase levels in mice with muscular dystrophy, including the classic citation by Morgan et al. (PMID: 7221187). Previous studies also reported varied CK activity level (Table R1). The differences for CK values of ~1000 in our wildtype mice from that in Morgan et al. paper may be attributed to different experimental and normalization methods. In addition, Morgan et al. study used Bar Harbor 129 ReJ strain that might be also different from C57/B6 mice used by us. In our study, we measured the CK activity with methods adapted from Sanchez-Castro et al. study (PMID: 33633730).

Fig. R2 (related to Fig. S1d): Time-course measurement of creatine kinase activity in DMD mice

Table R1: CK activity results reported in previous study

Background CK(U/liter) References

1 WT 327.00±118.90 PMID: 28439558

Dmd exon23 carried nonsense mutation (mdx) 7985.00±1825.96

2 WT 318 PMID: 25123483

mdx 8834.60±3928.75

3 WT ~300 PMID: 26721683

mdx ~14000

4 WT ~250 PMID: 34698513

Mice with a 4-bp deletion at

Dmd exon 4 (DmdE4*) ~2500

5 WT 223.71±85.69 PMID: 31591596

mdx 4744.25±2430.92

6 WT ~3000 PMID: 30854433

Dmd exon44 deletion (Δ Ex44) ~70000

7 WT ~1000 PMID: 33633730

8 Normal mice 24-102 PMID: 7221187

Dystrophic mic 79-157

Data Presentation: We have clarified that data are presented as mean +/- s.d. as stated in the figure legend and methods section.

Revertant fibers (Fig. R3): The number of revertant fibers was quantified in the revised manuscript.

Fig. R3 (related to Fig. S1b): Revertant fibers analysis in DMD mice with different ages
A majority of comments has been addressed. The detailed description of the method for counting centrally nucleated fibres is not yet clear ("obtained by counting the number of intra-nuclear fibers"). It is unclear what intra-nuclear fibres are.

Histology Fig. 1: Severe fibrosis is not shown in the TA.

Response: Thanks for your comment. Through Sirius red and HE staining, we observed severe fibrosis in the diaphragm and anterior tibialis muscles. The H&E and Sirius red staining results of 8 DMD mice are presented as follow (Fig. R3), and representative results have been selected to replace Fig.1e.

Reviewer: The now selected image seems more fitting. Fine.

Fig. 1H: The mean and SEM of CK levels do not reflect the data points shown.

Response: Thanks for raising this issue. We have replotted the results (Fig. R4) as suggested to revise the mistake.

Response Reviewer#1: The mean appears correct now. The figure legend of the figure in the second submission describes, contrary to the original submission file, that the SD is plotted. However, we would expect the SD error bars to be much larger in this case because two values are higher than the other four. If the authors intend to show the SD, please replot the figure.

Response: We appreciate reviewer's feedback regarding the figure legend and the representation of standard deviation (SD) error bars. We have ensured that this number is represented accurately with appropriate SD error bars and redraw the graph to reflect the expected representation of SD. It was fixed as commented.

These SD error bars, seem more plausible. OK

Discuss differences between human and murine exon 50. In which aspect is the model humanized?

Response: Although the homology between mouse and human DMD genes is very high, there are still significant sequence differences near the splicing site (Fig. R5). To generate a genetically humanized DMD mouse model carrying specific human exon deletion mutations, we knocked in human exon 50 with flanking 200 bp sequences to replace mouse exons 50 and 51 in a single step. This will facilitate the study of gene-editing-mediated exon skipping therapy.

Fig. R5: Alignment of human and mouse exon 50 sequence. Human exon 50 (Yellow labeling), mouse exon 50 (Blue labeling), sgRNA (magenta line), Protospacer adjacent motif (Red box).

Response Reviewer#1: Is this Fig. S1a? We appreciate that the comparison of the mouse and human sequences is now included.

Response: We appreciate your attention to this detail and are pleased that this addition has been recognized.

OK

Fig. 3C: Please comment on the additional band in the control-panel. The band demonstrates that you see exon 51 KO and in addition exons 50 plus 51 KO.

Response: Thanks for raising this issue. We sequenced the unexpectedly appearing band and found that this sequence was generated by exon skipping. We speculate that it is generated by

spontaneous exon skipping in mice.

Response Reviewer#1: OK.

Response: We appreciate your attention to this detail and are pleased that this addition has been recognized.

OK

Fig. 3B shows an A>G conversion of 20% for ABE1 and 25% for ABE2. How do the authors then explain 80 – 100% RNA skipping efficiency? Are there bystander edits? Please provide a chromatogram of the edited and unedited splice donors sites.

Response: Thanks for your good comment. We also find the discrepancy between genome editing and RNA skipping efficiency intriguing. Previous study (PMID: 34698513 , PMID: 30854433) reported similar results with us, which might be due to different stability of edited and unedited transcripts or methodologic difference between DNA and RNA editing analysis. For bystander editing analysis, we provided the reads analysis presented as follow (Fig. R6) to show alleles with or without bystander editing events.

Fig. R6 : Deep-seq reads analysis of ABE1- and ABE2-edited DMD gene in DMD mice.

Response Reviewer#1: Bystander editing rates are now shown. The discrepancy between the editing rates and RNA skipping efficiency should be discussed in the manuscript?

Response: As suggested, we added discussion on the discrepancy between the editing rates and RNA skipping efficiency in the revised manuscript as follow.

“This discrepancy has been reported in previous studies (PMID: 26721684; PMID: 30166439; PMID: 29805845; PMID: 32892813; PMID: 34698513) as summarized below (Table R2). The discrepancy is likely due to the presence of multiple cell types within the muscle tissue, including endothelial cells, pericytes, macrophages, fibro-adipogenic progenitors, and potentially other cell types that are not yet well understood at this time (PMID: 29305000). Additionally, nonsense-mediated decay (NMD) may influence the abundance of non-edited cDNA products, resulting in an apparent higher proportion of edited cDNA (PMID: 27259145). It is worth noting that, given the multinucleated nature of many murine muscles, even editing a single nucleus in a cardiomyocyte or skeletal muscles would result in the entire multinucleated cell regaining dystrophin protein, leading to a higher number of Dys+ cells compared to the relatively modest DNA base editing efficiency (PMID: 27259145).”

Table R2: Efficiency of DNA editing, RNA skipping and restored Dys+ fibers reported in previous study

DNA	RNA	Dys+ fibers	WB	References
1	~2%	59%	67%	~8% PMID: 26721684
2	2~4%	73.19%	-	60% PMID: 30166439
3	5.4%	-	Estimated 60%	- PMID: 29805845
4	Not provided	10.3%+4.3%	~72%	~50% PMID: 32892813
5	3.9±1.3%	59.98±4.74%	86.9±10.1%	- PMID: 34698513

Thank you for the provided overview and for adding this point to your discussion. We think this allows readers who are not familiar with publications such as those listed to understand the conclusions of your manuscript.

Fig. 4B and D: Please show bars on the same scale from 1 to 100% (same for 5B and D)

Response: Thanks for the suggestion. The Fig. 4B and D, Fig. 5B and D have been modified as suggested (Fig. R7).

Response Reviewer#1: The adjustments of the scales of 4D and the figure now called 5C is good. Here, the scale is now 0-100% as suggested. For 4B and 5B, however, the visualisation is less clear than before and harder to compare with the color-coded heat-map. Please find a way to show bar graphs side by side.

Response: We have used side-by-side display of bar graphs as suggested to improve clarity and facilitate comparison (Fig. R4). Your input is valuable to us, and we are committed to addressing this issue in the revised submission.

Fig. R4 (related to Fig. 4b): Heatmap and bar graph show base editing efficiency for DMD gene

The graphs can be compared more easily now.

Also the following improvements should be made:

- A) Addition of a label of the Y-Axis in the graphs of 4B/5B (0-20 ---- unit missing).
- B) How come the editing efficiency now seems to have dropped to 20% as compared to the original submission.

Response: Thank you for the reviewer's feedback. We have replaced the heatmap graph with bar graph and added Y-axis label as commented. Editing efficiency results in the original submission was obtained by EditR analysis of sanger sequencing, which might be less precise than next-generation sequencing of amplicon that we used in the revised manuscript. The NGS analysis methods of editing efficiency have been thoroughly detailed in the methods section now.

The provided explanation to the editing efficiency difference is plausible. Sanger-sequencing is less precise than NGS.

In the original submission you described in the figure legend that "deep sequencing" was used to measure the efficiency and not Sanger as described here. However, we appreciate the use of the more precise NGS method in this submission file.

Now in the second submission:

Fig. R7: Genome editing and RNA skipping efficiency after intraperitoneal administration of AAV-ABE2. a, Genome editing efficiency heatmap in heart, DI and TA for control and treated mice. b, Percentage of RNA skipping in heart, DI and TA for control and treated mice. n = 3. Data are represented as mean \pm SD. Each dot represents an individual mouse.

Fig. 4C and 5C: The sizes of the PCR products are not comprehensible. Please list all primers in the Supplement or name, if a different marker was used. A scheme of the size of the expected PCR products would ease the reading of the manuscript.

Response: Thank you for your careful review and helpful suggestions. The sizes of the PCR products were added in Fig. 4C and 5C shown as follow (Fig. R8), and all primers were also provided in Supplementary Table S1 of the revised manuscript.

Fig. R8: RT-PCR products from muscle of DMD mice were analyzed by gel electrophoresis. 449 bp and 340 bp bands are from transcripts without and with exon 50 skipping respectively.

Response Reviewer#1: It is now easier to follow the sizes of the PCR products with labelled PCR product sizes and marker lanes.

Response: Your acknowledgment of the enhanced visibility is greatly appreciated.

OK

Fig. S3: Please describe how the off-target analysis was performed. How many mismatches were allowed? How were the eight sites selected? Which PAM-specification was used for the prediction. The (Cas)-OFFinder is not cited (Bae et al, 2014).

Response: Thank you for your comment. We have revised the method description for off-target analysis. Using the PAM sequence (5'-NG-3') of XCas9 3.7 (TLIKDIV SpCas9) from *Streptococcus pyogenes* for prediction on the Cas-offinder website, all 14 potential off-target sites with 3 mismatched positions are selected for evaluation. In the revised manuscript, we have included all 14 predicted off-target sites with 3 mismatches and analyzed them using deep sequencing, showing high on-target editing with sgRNA6 but undetectable off-target editing events as follow (Fig. R9). In addition, we have cited the relevant paper as suggested.

Fig. R9: Off-target analysis of ABE2-mediated base editing for DMD gene. a, Alignment of the top 14 off-target sites in human genome. The target adenine (A7) is colored red. b, Percentages of adenine editing in the all 14 off-target sites. Dots and bars represent biological replicates and data are presented as means \pm SD (n = 3).

Response Reviewer#1: It is now included how off-target sites were selected with transparent inclusion criteria. Please adjust the figure legend describing in line 959 the alignment of the top TEN off-targets. It is 14.

Response: Thanks for the good comments. We have revised the typos in the manuscript.

OK

There is no mentioning of skipping in the muscle stem cells, satellite cells.

Response: Thanks for raising this issue. Dystrophin is expressed in differentiated myofibers and activated muscle stem cells (PMCID: PMC4839960). After intravenous injection of DMD mice with AAV-ABE2 for 6 weeks, we dissected the tibialis anterior muscle and isolated satellite cells for deep sequencing and RT-PCR analysis. The results showed that satellite cells have comparable A-to-G editing efficiency of up to 30% as the tibialis anterior muscle (Fig. R10). However, there is a low amount of mouse satellite cells, making it difficult to extract RNA and perform effective gel electrophoresis analysis of RNA skipping rate. Previous studies have reported that by using adeno-associated virus (AAV) to deliver gene editing tools targeting the DMD mutation region, it can be effectively delivered to muscle stem cells, allowing for gene editing and restoring their normal differentiation and renewal functions (PMID: 36995603, PMC4924477). Indeed, we think that your suggestion is very important and have taken it seriously in our lab to optimize satellite cells protocol for RNA skipping evaluation and hopefully get it done in another study.

Fig. R10: Gene editing analysis in satellite cells of DMD mice. a, Satellite cells of DMD mice treated with AAV-ABE2 showed PAX7 expression. b, Gene editing efficiency measured with deep-seq for satellite cells of DMD mice treated with AAV-ABE2. c, Deep-seq reads analysis for ABE2-edited DMD gene in satellite cells.

Response Reviewer#1: How were satellite cells isolated? Please provide the protocol in the Methods section. Have other myogenic markers been stained? The immunofluorescent picture that is provided does not clearly show nuclei positive for Pax7. The magnification is insufficient and the morphology of the cells do not fit.

Response: The protocol of satellite cells isolation was provided in the methods section as follow. We have addressed the concern by incorporating the use of both Pax3 and Pax7 antibodies (Fig. R5). During the satellite cell isolation and culture process, the cells exhibit a predominantly round

morphology, and gradually transition to a spindle-shaped morphology thereafter. The manuscript now includes the staining results of satellite cells after 4 days of isolation.

“The hindlimb and forelimb skeletal muscles were removed after the mice were euthanized. The muscles were washed twice with Dulbecco’s phosphate-buffered saline (DPBS) (Thermo Fisher Scientific) and finely chopped. Subsequently, they were digested using a 0.2% Collagenase type 2 (Gibco) solution for 60 minutes in a shaking water bath at 37°C. A second digestion was performed with a solution consisting of 0.2% Collagenase type 2 and 0.4% Dispase (GIBCO) in Rinsing media for 30 minutes in a shaking water bath at 37°C. The digested tissue was then passed through a 40mm filter to collect the filtrate. In order to increase the probability of available satellite cells, a purification process was conducted by wall sticking screening twice, each time for 1 hour. The prepared cell suspension was then inoculated into ECM-coated (Sigma) coverslips. The satellite cells were cultured in growth medium (Ham’s F10, 10% fetal bovine serum) supplemented with fibroblast growth factor (FGF) (Gibco) at 37°C in a humidified incubator with 5% CO₂.”

Fig. R5 (related to Fig. S10a): Immunostaining results for isolated satellite cells

We do not find it likely that all satellite cells are double positive for Pax3 and Pax7. Please remove R5 from the manuscript. The antibodies used for staining are not listed in the material & method section.

How long does the effect of skipping last? Was six weeks after editing the only timepoint that was checked? Please clarify how i.v. delivery was performed: retro-orbital or via tail vein?

Response: To examine the long-term therapeutic effect of ABE2 treatment, DMD mice with or without ABE2 administration were monitored for 10 months and then euthanized to analyze muscle tissues. Our results revealed durable therapeutic efficacy of ABE2 administration for 10 months, which are added in the revised manuscript presented as follow (Fig. R11).

Fig. R11: Intravenous delivery of ABE system efficiency rescues dystrophin expression and muscle function in humanized DMD mice. a, Schematic of intravenous administration of ABE2 particles. Tissues were collected for genomic DNA, RNA, immunoblotting and immunofluorescence experiments at 6 weeks (n=3) and 10 months (n=6) after treatment. Black arrows indicate time points for tissue collection after IV injection. b, Measurement by deep sequencing of splicing site editing efficiency in TA, DI, and heart after systemic delivery of ABE2. RT-PCR products from muscle of DMDΔmE5051,KIhE50/Y mice were analyzed by deep sequencing. c, RNA exon-skipping efficiency. d, Immunohistochemistry for dystrophin in TA, DI, and heart of DMDΔmE5051,KIhE50/Y mice was performed 6 weeks or 10 months after intravenous injection. Dystrophin is shown in green. Scale bar, 200 μm. e-f, Western blot analysis shows restoration of dystrophin expression in the TA, DI, and heart of DMDΔmE5051,KIhE50/Y mice 6 weeks or 10 months after injection. Dilutions of protein extract from WT mice were used to standardize dystrophin expression (10%, 25%, and 50%). Vinculin was used as the loading control. Forelimb grip strength (g) and rotarod rod performance (h) were measured two days in WT, and DMDΔmE5051,KIhE50/Y mice treated without or with ABE2 particles. i, The remaining strength was measured during 10 repetitions at 10-second intervals. Dots and bars represent biological replicates and data are presented mean ± SD. Significance is indicated by asterisks and was determined in Fig. 5g, h using unpaired two-tailed Student’s t test or in Fig. 5i using ANOVA multiple comparison test.

Response Reviewer#1: The 10 month timepoint is a valuable addition to the manuscript.

Response: We appreciate the reviewer’s comments on the significance of our findings.

OK

The claims in the discussion should be down-sized. It has not been examined whether gene editing byproducts were produced. Also, whether this is a safe alternative to existing strategies is also questionable. The topic of off-target analysis is discussed only very briefly. The specific ABE that was used is not discussed. The very similar paper of Chemello et al. 2021; DOI: 10.1126/sciadv.abg4910 has not been cited. The major difference to the paper by Chemello et al. is the fact that there was no humanized exon 50 in the Chemello-paper.

Response Reviewer#1: The Chemello paper is now mentioned including an explanation of what is unique to this manuscript. The sentence that the ABE strategy is safe was downsized.

Response: Thanks for helping improve our study. We have revised our manuscript as suggested and cited relevant papers.

OK

Minor:

Remove Company names from the legend

Response: Thank you for your careful review and helpful suggestions. Company names in the legend were removed as suggested.

Fig. 4F: Please add a legend to the columns either in the figure itself or in the legend text.

Response: The legend text has been added to the figures of our revised manuscript.

Fig. R12: Immunohistochemistry for dystrophin in TA, DI, and heart of DMD mice was performed 6 weeks after IP injection. Dystrophin is shown in green. Scale bar, 200 μ m.

Fig. S6: The legend does not correspond to the figure.

Response: We felt sorry for the mistake. The manuscript was revised as commented.

Response Reviewer#1: Ok. The legend of the now called Figure S7B still does not seem to reflect what is shown in the figure. Please explain why editing efficiency can be higher than 100%. also in S3:

Response: Thank you for the keen observation by reviewer#1. The legend of Fig. S7b was revised to reflect what is shown in the figure. For the editing efficiency presented in Fig. S3, we would like to clarify that the analysis was performed using relative editing efficiency and we normalized the data of intein-split ABE using the full-length ABE as reference and assigning mean value of full-length ABE to 100%. We have revised the sentence in the manuscript.

OK

Reviewer #2:

Remarks to the Author:

Reviewer #4:

Remarks to the Author:

The manuscript has benefitted from the revision work. Authors now have acknowledged the pioneering study of Chemello et al. who published a technically similar approach with the

difference that exon 50 was not humanized. Thus, the main novelty is the new mouse model and the detailed analysis of an AAV-mediated adenine base editing therapeutic strategy in a short- and long-term study.

Nevertheless, there are still limitations.

1. It is still unclear to which extent the study involves randomization and blinding. This reviewer thinks that blinding is of decisive importance in particular when using small animal numbers to rule out any potential bias.
2. Animal numbers in some experiments are still low. For example, quantitation of dystrophin expression in mice upon local injection is based on $n=3$ (Fig. 3h, i).
3. Statistical tests should be doublechecked as authors apply a Student's t test for comparison of three groups (see Fig. 4g, h and Fig. 5f,g). An Anova would be more appropriate as in Fig. 4i and Fig. 5h (line 595/624).
4. The statement regarding the animal experiments remains still unclear (line 703). What does „on a regular basis“ mean in line 703? Was there an individual assessment of the animal ethics of this research project? Can authors provide an individual number of the ethics assessment/permission?
5. Western blot for reviewers show only narrow sections of the bands of interest. No chance to assess quality of the original blots.

Reviewers' comments:

We are grateful for the positive feedback and constructive suggestions provided by the reviewers. Your valuable insights have offered valuable guidance and inspiration for our research work. We will carefully incorporate the recommendations that you have put forth, in order to further enhance the academic rigorousness and research quality of the manuscript. Our responses were highlighted in blue in response letter below and revisions in manuscript were marked in yellow.

REVIEWER COMMENTS

Reviewer #1 (Remarks to the Author):

We sincerely thank for your insightful feedback and constructive suggestions. We have addressed several of the issues highlighted (Red) in your comments and carefully responded to the remaining concerns for individual discussion, to ensure clear communication and thorough review.

Has the new mouse model been characterized before or after the age of 8 weeks (the age where analyses were performed, legend Fig. 1). Please comment on the reasons why this time point was chosen. The presence and abundance of revertant fibers should be commented on.

Response: Thank you for the good comment. We have performed phenotypic characterization of the humanized mouse model at 2-week, 8-week and 24-week. We only showed the results of 8-week DMD mice in Fig.1. In the revised manuscript, we have provided all results shown as follow (Fig. R1) for comprehensive phenotypic characterization of DMD mice at different time points. Briefly, the muscular histology of the DMD mice showed the inward migration of cell nuclei in all the tested muscle tissues occurred as early as 2 weeks old (Fig. R1a). As the mice grew older (8 weeks and 24 weeks old), the populations of centrally nucleated fibers (CNFs) significantly increased and the inflammatory cell infiltration also appeared at 2 weeks of age, with the increasing severity over time (Fig. R1b). The appearance of creatine kinase (CK) in blood has been considered as a biochemical marker of muscle necrosis. Consistent with dystrophin deficient DMD patients, serum CK activity of this humanized DMD mice had a dramatic elevation at 2 weeks of age when compared to the wildtype (WT) controls, indicating the severity of muscle damage (Fig. R1c). Moreover, the DMD mice exhibited significant reduction in muscle strength as early as 6 weeks old when compared to the age-matched WT mice (Fig. R1d), recapitulating progressive muscle weakness of DMD patients. Therefore, 8-week-old DMD mice were chosen for evaluation. In the humanized DMD mice, we detect very few revertant fibers that are difficult to be distinguished from background antibody staining signal (Fig. R2).

Response Reviewer#1: The figure is labeled S1 in the manuscript, not R1. The additional

results are important. The alignment of mouse and human sequence is helpful. The yellow arrowheads in S1b occlude parts of the sections and should be removed. The histology does not match the bar graphs (S1c). If 80% central nuclei are indicated in the bar graph then they should be present in histology as well. Please select representative images and indicate how many fibers were counted for analyzing the centrally nucleated fibers. Heart muscle fibers, physiologically, have a single (rarely two) centrally located nucleus. If nothing else is pointed out in the histologically features, the histology of heart sections can be removed. Fig S1d: CK levels: There is quite a literature on creatine kinase levels (U/l) in mice \pm muscular dystrophy. One classical citation is: Morgan et al. Res Commun Chem Pathol Pharmacol PMID: 7221187. The level in wildtype mice do not exceed 150-200 U/l. Please comment on values of 1000 and above. The authors state in the figure below that the data are presented as mean \pm SEM. The now submitted manuscript states it is the mean \pm SD. Which one is true? Please quantify revertant fibers.

Response: We appreciate the valuable feedback provided by the reviewer. In response to the concerns raised, we have made the following revisions to the manuscript:

Figure Labeling: We acknowledge the oversight in labeling of the Fig. R1 instead of Fig. S1 in the manuscript. We recognize the importance of the additional results and have ensured that they are appropriately highlighted in the revised manuscript.

Fig. S1b: We have removed the yellow arrowheads in Fig. S1b to ensure that the sections are clearly visible in the revised figure.

Histology and Bar Graphs (Fig. R1): We have addressed the discrepancy between the histology and the bar graphs in Fig. S1c by selecting representative images and providing a detailed description of the fiber counting method to ensure accuracy in the analysis of centrally nucleated fibers as follow. Additionally, the histology of heart sections was removed as suggested.

“...The percentage of central nucleation fibers is obtained by counting the number of intra-nuclear fibers in whole-muscle scanning of GA, DI, and QA muscles....”

Fig. R1 (related to Fig. S1c, d): Histological analysis of DMD mice (a) and centrally nucleated fibers statistics (b)

CK Levels (Fig. R2): We have conducted a thorough literature review (Table R1) on creatine kinase levels in mice with muscular dystrophy, including the classic citation by Morgan et al. (PMID: 7221187). Previous studies also reported varied CK activity level (Table R1). The differences for CK values of \sim 1000 in our wildtype mice from that in Morgan et al. paper may be attributed to different experimental and normalization methods. In addition, Morgan et al. study used Bar Harbor 129 ReJ strain that might be also different from C57/B6 mice used by us. In our study, we measured the CK activity with methods adapted from Sanchez-Castro et al. study (PMID: 33633730).

Fig. R2 (related to Fig. S1d): Time-course measurement of creatine kinase activity in DMD

mice

Table R1: CK activity results reported in previous study

Background CK(U/liter) References

1 WT 327.00±118.90 PMID: 28439558

Dmd exon23 carried nonsense mutation (mdx) 7985.00±1825.96

2 WT 318 PMID: 25123483

mdx 8834.60±3928.75

3 WT ~300 PMID: 26721683

mdx ~14000

4 WT ~250 PMID: 34698513

Mice with a 4-bp deletion at

Dmd exon 4 (DmdE4*) ~2500

5 WT 223.71±85.69 PMID: 31591596

mdx 4744.25±2430.92

6 WT ~3000 PMID: 30854433

Dmd exon44 deletion (Δ Ex44) ~70000

7 WT ~1000 PMID: 33633730

8 Normal mice 24-102 PMID: 7221187

Dystrophic mic 79-157

Data Presentation: We have clarified that data are presented as mean +/- s.d. as stated in the figure legend and methods section.

Revertant fibers (Fig. R3): The number of revertant fibers was quantified in the revised manuscript.

Fig. R3 (related to Fig. S1b): Revertant fibers analysis in DMD mice with different ages

A majority of comments has been addressed. The detailed description of the method for counting centrally nucleated fibres is not yet clear ("obtained by counting the number of intra-nuclear fibers"). It is unclear what intra-nuclear fibres are.

Response: Thank you for your feedback. We appreciate your attention to the clarity of our manuscript. In response to your comments regarding the description of our method for quantifying centrally nucleated fibers (CNFs), we acknowledge that our previous terminology may have caused confusion. To clarify, we have revised the description of our methodology as follows:

"To quantify centrally nucleated fibers (CNFs), the following methodology is employed: Slides stained with H&E staining were examined under a light microscope at a magnification of 20x. Each muscle fiber was evaluated to determine whether the nucleus was centrally positioned rather than peripherally. A minimum of 500 muscle fibers were counted per sample to ensure statistical reliability. Fibers were classified as centrally nucleated if the nucleus was located within the central third of the fiber's cross-sectional

area. The percentage of CNFs was calculated by dividing the number of centrally nucleated fibers by the total number of fibers counted and then multiplying by 100 to obtain a percentage.”

Histology Fig. 1: Severe fibrosis is not shown in the TA.

Response: Thanks for your comment. Through Sirius red and HE staining, we observed severe fibrosis in the diaphragm and anterior tibialis muscles. The H&E and Sirius red staining results of 8 DMD mice are presented as follow (Fig. R3), and representative results have been selected to replace Fig.1e.

Reviewer: The now selected image seems more fitting. Fine.

Response: Thank you for your feedback.

Fig. 1H: The mean and SEM of CK levels do not reflect the data points shown.

Response: Thanks for raising this issue. We have replotted the results (Fig. R4) as suggested to revise the mistake.

Response Reviewer#1: The mean appears correct now. The figure legend of the figure in the second submission describes, contrary to the original submission file, that the SD is plotted. However, we would expect the SD error bars to be much larger in this case because two values are higher than the other four. If the authors intend to show the SD, please replot the figure.

Response: We appreciate reviewer’s feedback regarding the figure legend and the representation of standard deviation (SD) error bars. We have ensured that this number is represented accurately with appropriate SD error bars and redraw the graph to reflect the expected representation of SD. It was fixed as commented.

These SD error bars, seem more plausible. OK

Discuss differences between human and murine exon 50. In which aspect is the model humanized?

Response: Although the homology between mouse and human DMD genes is very high, there are still significant sequence differences near the splicing site (Fig. R5). To generate a genetically humanized DMD mouse model carrying specific human exon deletion mutations, we knocked in human exon 50 with flanking 200 bp sequences to replace mouse exons 50 and 51 in a single step. This will facilitate the study of gene-editing-mediated exon skipping therapy.

Fig. R5: Alignment of human and mouse exon 50 sequence. Human exon 50 (Yellow labeling), mouse exon 50 (Blue labeling), sgRNA (magenta line), Protospacer adjacent

motif (Red box).

Response Reviewer#1: Is this Fig. S1a? We appreciate that the comparison of the mouse and human sequences is now included.

Response: We appreciate your attention to this detail and are pleased that this addition has been recognized.

OK

Fig. 3C: Please comment on the additional band in the control-panel. The band demonstrates that you see exon 51 KO and in addition exons 50 plus 51 KO.

Response: Thanks for raising this issue. We sequenced the unexpectedly appearing band and found that this sequence was generated by exon skipping. We speculate that it is generated by spontaneous exon skipping in mice.

Response Reviewer#1: OK.

Response: We appreciate your attention to this detail and are pleased that this addition has been recognized.

OK

Fig. 3B shows an A>G conversion of 20% for ABE1 and 25% for ABE2. How do the authors then explain 80 – 100% RNA skipping efficiency? Are there bystander edits? Please provide a chromatogram of the edited and unedited splice donors sites.

Response: Thanks for your good comment. We also find the discrepancy between genome editing and RNA skipping efficiency intriguing. Previous study (PMID: 34698513 , PMID: 30854433) reported similar results with us, which might be due to different stability of edited and unedited transcripts or methodologic difference between DNA and RNA editing analysis. For bystander editing analysis, we provided the reads analysis presented as follow (Fig. R6) to show alleles with or without bystander editing events.

Fig. R6: Deep-seq reads analysis of ABE1- and ABE2-edited DMD gene in DMD mice.

Response Reviewer#1: Bystander editing rates are now shown. The discrepancy between the editing rates and RNA skipping efficiency should be discussed in the manuscript?

Response: As suggested, we added discussion on the discrepancy between the editing rates and RNA skipping efficiency in the revised manuscript as follow.

“This discrepancy has been reported in previous studies (PMID: 26721684; PMID:

30166439; PMID: 29805845; PMID: 32892813; PMID: 34698513) as summarized below (Table R2). The discrepancy is likely due to the presence of multiple cell types within the muscle tissue, including endothelial cells, pericytes, macrophages, fibro-adipogenic progenitors, and potentially other cell types that are not yet well understood at this time (PMID: 29305000). Additionally, nonsense-mediated decay (NMD) may influence the abundance of non-edited cDNA products, resulting in an apparent higher proportion of edited cDNA (PMID: 27259145). It is worth noting that, given the multinucleated nature of many murine muscles, even editing a single nucleus in a cardiomyocyte or skeletal muscles would result in the entire multinucleated cell regaining dystrophin protein, leading to a higher number of Dys+ cells compared to the relatively modest DNA base editing efficiency (PMID: 27259145).”

Table R2: Efficiency of DNA editing, RNA skipping and restored Dys+ fibers reported in previous study

DNA RNA Dys+ fibers WB References

1 ~2% 59% 67% ~8% PMID: 26721684

2 2~4% 73.19% - 60% PMID: 30166439

3 5.4% - Estimated 60% - PMID: 29805845

4 Not provided 10.3%+4.3% ~72% ~50% PMID: 32892813

5 3.9±1.3% 59.98±4.74% 86.9±10.1% - PMID: 34698513

Thank you for the provided overview and for adding this point to your discussion. We think this allows readers who are not familiar with publications such as those listed to understand the conclusions of your manuscript.

Response: We appreciate your acknowledgment of our efforts to improve the manuscript and thank you for guiding us to better serve the needs of all readers. We believe that these adjustments will significantly enhance the manuscript's impact by making it more inclusive and informative for both specialists and non-specialists alike.

Fig. 4B and D: Please show bars on the same scale from 1 to 100% (same for 5B and D)

Response: Thanks for the suggestion. The Fig. 4B and D, Fig. 5B and D have been modified as suggested (Fig. R7).

Response Reviewer#1: The adjustments of the scales of 4D and the figure now called 5C is good. Here, the scale is now 0-100% as suggested. For 4B and 5B, however, the visualisation is less clear than before and harder to compare with the color-coded heatmap. Please find a way to show bar graphs side by side.

Response: We have used side-by-side display of bar graphs as suggested to improve clarity and facilitate comparison (Fig. R4). Your input is valuable to us, and we are committed to addressing this issue in the revised submission.

Fig. R4 (related to Fig. 4b): Heatmap and bar graph show base editing efficiency for DMD

gene

The graphs can be compared more easily now.

Response: Thank you for your feedback regarding the improvements to the graphs in our manuscript.

Also the following improvements should be made:

- A) Addition of a label of the Y-Axis in the graphs of 4B/5B (0-20 ---- unit missing).
- B) How come the editing efficiency now seems to have dropped to 20% as compared to the original submission.

Response: Thank you for the reviewer's feedback. We have replaced the heatmap graph with bar graph and added Y-axis label as commented. Editing efficiency results in the original submission was obtained by EditR analysis of sanger sequencing, which might be less precise than next-generation sequencing of amplicon that we used in the revised manuscript. The NGS analysis methods of editing efficiency have been thoroughly detailed in the methods section now.

The provided explanation to the editing efficiency difference is plausible. Sanger-sequencing is less precise than NGS.

In the original submission you described in the figure legend that "deep sequencing" was used to measure the efficiency and not Sanger as described here. However, we appreciate the use of the more precise NGS method in this submission file.

Response: Thank you for acknowledging the plausibility of our explanation regarding the observed differences in editing efficiency between Sanger sequencing and next-generation sequencing (NGS). We appreciate your affirmation of the points raised in our manuscript concerning the comparative precision of these sequencing methodologies.

In the original submission, the term "deep sequencing" was indeed mentioned in the figure legend, which may have led to some confusion. We apologize for any ambiguity this might have caused. In the revised manuscript, we employed NGS, which is more precise than Sanger sequencing, to measure the editing efficiency. This change was made to ensure a more accurate and reliable analysis of the editing outcomes.

Now in the second submission:

Fig. R7: Genome editing and RNA skipping efficiency after intraperitoneal administration of AAV-ABE2. a, Genome editing efficiency heatmap in heart, DI and TA for control and treated mice. b, Percentage of RNA skipping in heart, DI and TA for control and treated mice. n = 3. Data are represented as mean \pm SD. Each dot represents an individual mouse.

Fig. 4C and 5C: The sizes of the PCR products are not comprehensible. Please list all primers in the Supplement or name, if a different marker was used. A scheme of the size of the expected PCR products would ease the reading of the manuscript.

Response: Thank you for your careful review and helpful suggestions. The sizes of the PCR products were added in Fig. 4C and 5C shown as follow (Fig. R8), and all primers were also provided in Supplementary Table S1 of the revised manuscript.

Fig. R8: RT-PCR products from muscle of DMD mice were analyzed by gel electrophoresis. 449 bp and 340 bp bands are from transcripts without and with exon 50 skipping respectively.

Response Reviewer#1: It is now easier to follow the sizes of the PCR products with labelled PCR product sizes and marker lanes.

Response: Your acknowledgment of the enhanced visibility is greatly appreciated.

OK

Fig. S3: Please describe how the off-target analysis was performed. How many mismatches were allowed? How were the eight sites selected? Which PAM-specification was used for the prediction. The (Cas)-OFFinder is not cited (Bae et al, 2014).

Response: Thank you for your comment. We have revised the method description for off-target analysis. Using the PAM sequence (5'-NG-3') of XCas9 3.7 (TLIKDIV SpCas9) from *Streptococcus pyogenes* for prediction on the Cas-offinder website, all 14 potential off-target sites with 3 mismatched positions are selected for evaluation. In the revised manuscript, we have included all 14 predicted off-target sites with 3 mismatches and analyzed them using deep sequencing, showing high on-target editing with sgRNA6 but undetectable off-target editing events as follow (Fig. R9). In addition, we have cited the relevant paper as suggested.

Fig. R9: Off-target analysis of ABE2-mediated base editing for DMD gene. a, Alignment of the top 14 off-target sites in human genome. The target adenine (A7) is colored red. b, Percentages of adenine editing in the all 14 off-target sites. Dots and bars represent biological replicates and data are presented as means \pm SD (n = 3).

Response Reviewer#1: It is now included how off-target sites were selected with transparent inclusion criteria. Please adjust the figure legend describing in line 959 the alignment of the top TEN off-targets. It is 14.

Response: Thanks for the good comments. We have revised the typos in the manuscript.

OK

There is no mentioning of skipping in the muscle stem cells, satellite cells.

Response: Thanks for raising this issue. Dystrophin is expressed in differentiated myofibers and activated muscle stem cells (PMCID: PMC4839960). After intravenous injection of DMD mice with AAV-ABE2 for 6 weeks, we dissected the tibialis anterior muscle

and isolated satellite cells for deep sequencing and RT-PCR analysis. The results showed that satellite cells have comparable A-to-G editing efficiency of up to 30% as the tibialis anterior muscle (Fig. R10). However, there is a low amount of mouse satellite cells, making it difficult to extract RNA and perform effective gel electrophoresis analysis of RNA skipping rate. Previous studies have reported that by using adeno-associated virus (AAV) to deliver gene editing tools targeting the DMD mutation region, it can be effectively delivered to muscle stem cells, allowing for gene editing and restoring their normal differentiation and renewal functions (PMID: 36995603, PMC4924477). Indeed, we think that your suggestion is very important and have taken it seriously in our lab to optimize satellite cells protocol for RNA skipping evaluation and hopefully get it done in another study.

Fig. R10: Gene editing analysis in satellite cells of DMD mice. a, Satellite cells of DMD mice treated with AAV-ABE2 showed PAX7 expression. b, Gene editing efficiency measured with deep-seq for satellite cells of DMD mice treated with AAV-ABE2. c, Deep-seq reads analysis for ABE2-edited DMD gene in satellite cells.

Response Reviewer#1: How were satellite cells isolated? Please provide the protocol in the Methods section. Have other myogenic markers been stained? The immunofluorescent picture that is provided does not clearly show nuclei positive for Pax7. The magnification is insufficient and the morphology of the cells do not fit.

Response: The protocol of satellite cells isolation was provided in the methods section as follow. We have addressed the concern by incorporating the use of both Pax3 and Pax7 antibodies (Fig. R5). During the satellite cell isolation and culture process, the cells exhibit a predominantly round morphology, and gradually transition to a spindle-shaped morphology thereafter. The manuscript now includes the staining results of satellite cells after 4 days of isolation.

“The hindlimb and forelimb skeletal muscles were removed after the mice were euthanized. The muscles were washed twice with Dulbecco’s phosphate-buffered saline (DPBS) (Thermo Fisher Scientific) and finely chopped. Subsequently, they were digested using a 0.2% Collagenase type 2 (Gibco) solution for 60 minutes in a shaking water bath at 37°C. A second digestion was performed with a solution consisting of 0.2% Collagenase type 2 and 0.4% Dispase (GIBCO) in Rinsing media for 30 minutes in a shaking water bath at 37°C. The digested tissue was then passed through a 40mm filter to collect the filtrate. In order to increase the probability of available satellite cells, a purification process was conducted by wall sticking screening twice, each time for 1 hour. The prepared cell suspension was then inoculated into ECM-coated (Sigma) coverslips. The satellite cells were cultured in growth medium (Ham’s F10, 10% fetal bovine serum) supplemented with fibroblast growth factor (FGF) (Gibco) at 37°C in a humidified incubator with 5% CO₂.”

Fig. R5 (related to Fig. S10a): Immunostaining results for isolated satellite cells

We do not find it likely that all satellite cells are double positive for Pax3 and Pax7. Please remove R5 from the manuscript. The antibodies used for staining are not listed in the material & method section.

Response: Thank you for your comments and the concerns raised regarding our characterization of satellite cells as double positive for Pax3 and Pax7. In response to your suggestion, we have revised our immunofluorescence methodology to adjust the brightness settings, enhancing the visualization of Pax3 and Pax7 expression. This adjustment allows for more detailed observations and a better representation of the consistency of marker expression across satellite cell populations (Fig. R1).

Additionally, we have removed R5 from our manuscript as suggested.

Furthermore, we recognize that the omission of specific antibody details in the Materials and Methods section indeed exists. We have now included a detailed list of all antibodies used, completed with their sources and dilutions, to ensure reproducibility and provide full transparency of our experimental procedures.

“Subsequently, the cells were incubated overnight at 4°C with primary antibodies against PAX7 (DSHB, 042349, 1:500) and PAX3 (Beyotime, AF7686, 1:500) diluted appropriately in PBS.”

Fig.R1 Pax3 and Pax7 immunostaining results for satellite cells (muscle stem cells) isolated from ABE2-treated DMD^{ΔmE5051,KlhE50/Y} mice

How long does the effect of skipping last? Was six weeks after editing the only timepoint that was checked? Please clarify how i.v. delivery was performed: retro-orbital or via tail vein?

Response: To examine the long-term therapeutic effect of ABE2 treatment, DMD mice with or without ABE2 administration were monitored for 10 months and then euthanized to analyze muscle tissues. Our results revealed durable therapeutic efficacy of ABE2 administration for 10 months, which are added in the revised manuscript presented as

follow (Fig. R11).

Fig. R11: Intravenous delivery of ABE system efficiency rescues dystrophin expression and muscle function in humanized DMD mice. a, Schematic of intravenous administration of ABE2 particles. Tissues were collected for genomic DNA, RNA, immunoblotting and immunofluorescence experiments at 6 weeks (n=3) and 10 months (n=6) after treatment. Black arrows indicate time points for tissue collection after IV injection. b, Measurement by deep sequencing of splicing site editing efficiency in TA, DI, and heart after systemic delivery of ABE2. RT-PCR products from muscle of DMD Δ mE5051,KIhE50/Y mice were analyzed by deep sequencing. c, RNA exon-skipping efficiency. d, Immunohistochemistry for dystrophin in TA, DI, and heart of DMD Δ mE5051,KIhE50/Y mice was performed 6 weeks or 10 months after intravenous injection. Dystrophin is shown in green. Scale bar, 200 μ m. e-f, Western blot analysis shows restoration of dystrophin expression in the TA, DI, and heart of DMD Δ mE5051,KIhE50/Y mice 6 weeks or 10 months after injection. Dilutions of protein extract from WT mice were used to standardize dystrophin expression (10%, 25%, and 50%). Vinculin was used as the loading control. Forelimb grip strength (g) and rotarod rod performance (h) were measured two days in WT, and DMD Δ mE5051,KIhE50/Y mice treated without or with ABE2 particles. i, The remaining strength was measured during 10 repetitions at 10-second intervals. Dots and bars represent biological replicates and data are presented mean \pm SD. Significance is indicated by asterisks and was determined in Fig. 5g, h using unpaired two-tailed Student's t test or in Fig. 5i using ANOVA multiple comparison test.

Response Reviewer#1: The 10 month timepoint is a valuable addition to the manuscript.

Response: We appreciate the reviewer's comments on the significance of our findings.

OK

The claims in the discussion should be down-sized. It has not been examined whether gene editing byproducts were produced. Also, whether this is a safe alternative to existing strategies is also questionable. The topic of off-target analysis is discussed only very briefly. The specific ABE that was used is not discussed. The very similar paper of Chemello et al. 2021; DOI: 10.1126/sciadv.abg4910 has not been cited. The major difference to the paper by Chemello et al. is the fact that there was no humanized exon 50 in the Chemello-paper.

Response Reviewer#1: The Chemello paper is now mentioned including an explanation of what is unique to this manuscript. The sentence that the ABE strategy is safe was downsized.

Response: Thanks for helping improve our study. We have revised our manuscript as suggested and cited relevant papers.

OK

Minor:

Remove Company names from the legend

Response: Thank you for your careful review and helpful suggestions. Company names in the legend were removed as suggested.

Fig. 4F: Please add a legend to the columns either in the figure itself or in the legend text.

Response: The legend text has been added to the figures of our revised manuscript.

Fig. R12: Immunohistochemistry for dystrophin in TA, DI, and heart of DMD mice was performed 6 weeks after IP injection. Dystrophin is shown in green. Scale bar, 200 μm .

Fig. S6: The legend does not correspond to the figure.

Response: We felt sorry for the mistake. The manuscript was revised as commented.

Response Reviewer#1: Ok. The legend of the now called Figure S7B still does not seem to reflect what is shown in the figure. Please explain why editing efficiency can be higher than 100%. also in S3:

Response: Thank you for the keen observation by reviewer#1. The legend of Fig. S7b was revised to reflect what is shown in the figure. For the editing efficiency presented in Fig. S3, we would like to clarify that the analysis was performed using relative editing efficiency and we normalized the data of intein-split ABE using the full-length ABE as reference and assigning mean value of full-length ABE to 100%. We have revised the sentence in the manuscript.

OK

Reviewer #2 (Remarks to the Author):

Reviewer #4 (Remarks to the Author):

The manuscript has benefitted from the revision work. Authors now have acknowledged the pioneering study of Chemello et al. who published a technically similar approach with the difference that exon 50 was not humanized. Thus, the main novelty is the new mouse model and the detailed analysis of an AAV-mediated adenine base editing therapeutic strategy in a short- and long-term study.

Response: We appreciate your highly professional comments and suggestions.

Nevertheless, there are still limitations.

1. It is still unclear to which extend the study involves randomization and blinding. This reviewer thinks that blinding is of decisive importance in particular when using small animal numbers to rule out any potential bias.

Response: Thank you for your insightful comments regarding the clarity of randomization and blinding in our study. We acknowledge the critical importance of these methodologies, especially when conducting experiments with small animal cohorts, to eliminate potential biases and enhance the validity of our results.

In response to your concerns, we have revised the manuscript to include a more detailed description of the randomization and blinding procedures employed in our study. Specifically, we randomly assigned DMD mice to either treatment or control groups to ensure an unbiased distribution. Furthermore, the researchers responsible for assessing the outcomes were blinded to the group allocations. This blinding was rigorously maintained throughout the data collection and analysis phases to prevent any subjective bias in interpreting the results.

We believe these enhancements in our methodology description will clarify the robustness of our experimental design and reinforce the reliability of our findings. Thank you once again for your constructive feedback, which has significantly contributed to improving the quality and integrity of our study.

2. Animal numbers in some experiments are still low. For example, quantitation of dystrophin expression in mice upon local injection is based on $n=3$ (Fig. 3h, i).

Response: Thank you for your comments regarding the sample size in our experiments, particularly related to the quantitation of dystrophin expression in mice as depicted in Fig. 3h and i. We acknowledge that a sample size of $n=3$ may appear limited.

While we recognize the importance of sufficient sample sizes for robust statistical analysis, it is essential to clarify that the choice of $n=3$ was driven by the preliminary nature of these experiments. The primary aim was to establish feasibility and to collect initial data to inform subsequent experiments on systemic treatment. It is important to note that each experiment was meticulously controlled, and despite the small sample size, the results achieved statistical significance. Furthermore, a sample size of $n=3$ for intramuscular injection is consistent with many DMD gene therapy studies (referenced in Table R1), indicating that our approach aligns with the standard practices in preliminary experimental research.

Table R1 The sample size for intramuscular injections reported in previous studies.

	Samples number	References
1	4	PMID: 26721683
2	3-6	PMID: 29224783
3	4	PMID: 29187645

4	3	PMID: 29702637
5	4	PMID: 28195574
6	5	PMID: 29730196
7	3	PMID: 31586095
8	3	PMID: 32462052
9	4	PMID: 32222157
10	3	PMID: 32892813
11	3	PMID: 33931459
12	4	PMID: 34509668
13	4	PMID: 37215149
14	3	PMID: 37637209
15	3	PMID: 37098587
16	4	PMID: 36512423

We appreciate your highlighting this critical point, which has enabled us to better elucidate the scope and implications of our findings. This feedback has prompted us to include a more detailed discussion in the manuscript about the limitations associated with our sample size and the preliminary nature of our results, while also outlining plans for subsequent studies with increased sample sizes. We believe that this additional context will help readers to better understand our research approach and the foundational nature of our initial findings.

3. Statistical tests should be doublechecked as authors apply a Student's t test for comparison of three groups (see Fig. 4g, h and Fig. 5f,g). An Anova would be more appropriate as in Fig. 4i and Fig. 5h (line 595/624).

Response: Thank you for your insightful comments regarding the statistical methods employed in our study. Upon reviewing your observations, we acknowledge the inappropriate application of Student's t-test for the comparison of three groups in Fig. 4g, h and 5f, g. We agree that an ANOVA is indeed more suitable for these analyses, as correctly implemented in Fig. 4 and 5.

We have revised our manuscript to correct this error and have now applied ANOVA to the relevant figures to ensure statistical accuracy and integrity. This adjustment provides a more appropriate analysis and strengthens the validity of our results. We appreciate your attention to details, which has significantly contributed to improving the quality of our research.

4. The statement regarding the animal experiments remains still unclear (line 703). What does „on a regular basis“ means in line 703? Was there an individual assessment of the animal ethics of this research project? Can authors provide an individual number of the ethics assessment/permission?

Response: Thank you for your comments requesting clarification on the statement regarding animal experiments noted in line 703 of our manuscript. By "on a regular basis," we refer to the annual review process conducted by the City Science and Technology

Commission. This yearly evaluation covers various aspects including facility operations, animal care, and ethical reviews pertaining to the use of experimental animals. Only after passing this comprehensive inspection is the seal stamped, thereby extending the validity of our animal facility license for another year.

Additionally, we confirmed that an individual assessment of the animal ethics for this specific research project was conducted and the ethical approval for this study was granted by our Institutional Animal Care and Use Committee (IACUC), under the approval number HGAF2021-008.

We hope that this response adequately addresses your concern and clarifies the procedures followed to ensure ethical compliance in our research.

5. Western blot for reviewers show only narrow sections of the bands of interest. No chance to assess quality of the original blots.

Response: We have addressed the concern raised by the reviewer regarding the limited view of the bands of interest in the Western blot images. Full scans of the original blots have been provided in the supplementary materials for a more comprehensive assessment of the blot quality. Thank you for bringing this to our attention, and we believe that the additional information will enhance the clarity and transparency of our results.